# INFER: Embedding Integration with Feature Refinement for Few-Shot Learning in VLMs

## Abstract

On general visual recognition tasks, CLIP has demonstrated remarkable few-shot performance by aligning image and text modalities. However, CLIP's *sole* reliance on the class (CLS) embedding for image representation limits its capacity to capture spatially fine-grained features, which are crucial for fine-grained classification tasks, where subtle differences (e.g., bird species, car models) depend on small, localized variations rather than just the overall object outline. To address this, we introduce INFER, a feature enhancement strategy for CLIP that enhances image embeddings with spatial information intelligently extracted from patch embeddings as well as the CLS embedding. To this end, INFER leverages attention heads to compute attention-weighted features of both the patch and CLS embeddings. The most informative heads for each class, identified by their alignment with class text embeddings, are selected to enrich the patch and CLS features, which are then integrated through a lightweight fusion module. In the few-shot learning paradigm, INFER establishes new SOTA performance, highlighting the underutilized potential of CLIP's internal attention mechanisms and providing a generalizable framework for patch-level enhancement in CLIP.

## 1 Introduction

The advent of vision-language models (VLMs) has transformed computer vision, enabling robust few-shot generalization across a wide range of tasks. In particular, Contrastive Language–Image Pre-training (CLIP) has emerged as a cornerstone, achieving remarkable performance in aligning visual and textual modalities through contrastive learning on large-scale image–text datasets Radford et al. (2021). Despite CLIP's impressive capabilities, an inherent limitation persists that hinders its effectiveness in real-world applications. CLIP's architecture relies exclusively on the class (CLS) embedding at the last layer of vision encoder for its final image representation, discarding the rich spatial and semantic information encoded in patch embeddings across multiple layers Dong et al. (2023); Zhou et al. (2022a). This design choice, while computationally efficient, discards fine-grained visual details that are crucial for distinguishing semantically similar classes. The problem is particularly pronounced in fine-grained recognition tasks, where rich spatial semantics are essential.

Studies in patch-inclusive representation learning have demonstrated the potential of leveraging patch-level information for image segmentation tasks. MaskCLIP Dong et al. (2023) pioneered extracting dense labels from CLIP by directly utilizing patch embeddings, while ZegFormer Ding et al. (2022) separated the class-agnostic grouping from segment-level classification, providing insights into how spatial information can be effectively incorporated into patches.

While patch-inclusive studies highlight the importance of spatial information, recent advances in vision transformer (ViT) interpretation and attention analysis reveal that different heads capture distinct types of visual information, ranging from low-level textures to high-level semantic relationships Zhang et al. (2022); Gandelsman et al. (2023). This observation suggests that treating all attention heads in CLIP equally can be suboptimal, since some heads contribute more effectively to specific classification tasks than others. Moreover, patch-to-patch attention patterns within transformer layers encode valuable spatial relationships and contextual dependencies that remain unexploited in CLIP's current formulation. This suggests that further work is needed to determine *which heads are most useful* and *what information they capture* to better exploit CLIP's internal representations for fine-grained classification.

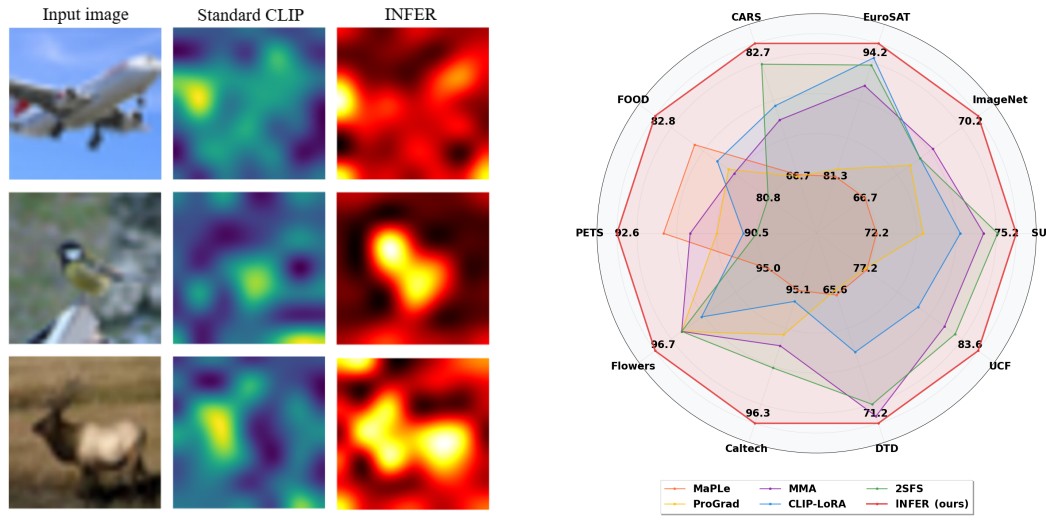

(a) Heatmaps of CLIP and INFER.

(b) Benchmark performance across datasets.

Figure 1: (a) The heatmaps of CLIP and INFER. (b) Classification accuracy on ten different datasets across diverse domains. INFER consistently outperforms all competing approaches, including the current SOTA method, 2SFS Farina et al. (2025).

Fine-grained classification poses a particularly challenging setting where subtle inter-class differences, such as variations in texture, shape, or local part configurations, are crucial for accurate recognition. Prompt-based adaptation has become a dominant paradigm for tuning VLMs in few-shot settings. Methods such as CoOp Zhou et al. (2022c), CoCoOp Zhou et al. (2022b), and their successors (e.g., PLOT++ Chen et al. (2022), MaPLe Khattak et al. (2023), ProGrad Zhu et al. (2023)) learn discrete or soft prompt vectors that condition the text encoder. These approaches often enhance generalization by introducing multiple prompts, aligning prompt updates with CLIP's gradients, or coupling prompts across modalities. Despite their effectiveness, prompt-based techniques still remain global in nature: they modify only the textual context while leaving visual embeddings unchanged. As a result, they often struggle to capture fine-grained spatial cues and can be sensitive to clutter or background noise.

Another line of work introduces lightweight modules or residual layers within the encoders to adapt representations. Adapter-based techniques such as MMA Yang et al. (2024) and classifier-oriented methods like TaskRes Yu et al. (2023) and LP+ Huang et al. (2024) improve alignment or discrimination by inserting trainable components at intermediate or final layers. Most recently, 2SFS Farina et al. (2025) fine-tunes only normalization layers in a two-stage schedule before training a classifier. These methods achieve strong performance with relatively few parameters but often require carefully staged optimization.

We introduce Embedding **IN**tegration with **FE**ature **R**efinement (INFER), a novel framework that improves CLIP's performance in recognition tasks. INFER avoids prompt learning and encoder modifications by adopting an embedding-integration strategy that combines the enriched CLS and patch embeddings through a fusion module. These embeddings are enhanced by our feature enhancement mechanism, yielding more discriminative and robust representations while preserving the frozen backbone. Figure 1 provides the qualitative and quantitative analysis of INFER. Figure 1(a) presents the heatmaps of standard CLIP and INFER, showing that INFER produces more spatially focused and semantically aligned activations by selectively enhancing CLS as well as patch features through informative attention heads. Figure 1(b) shows that INFER establishes new SOTA performance across diverse domains, outperforming all completing methods, including the current SOTA method, 2SFS Farina et al. (2025).

INFER operates on three key insights: (i) Effective fine-grained visual understanding requires both global context (captured by the CLS embedding) and local spatial relationships (encoded in patch embeddings), (ii) Selectively enhancing the spatial and semantic information in patch and CLS

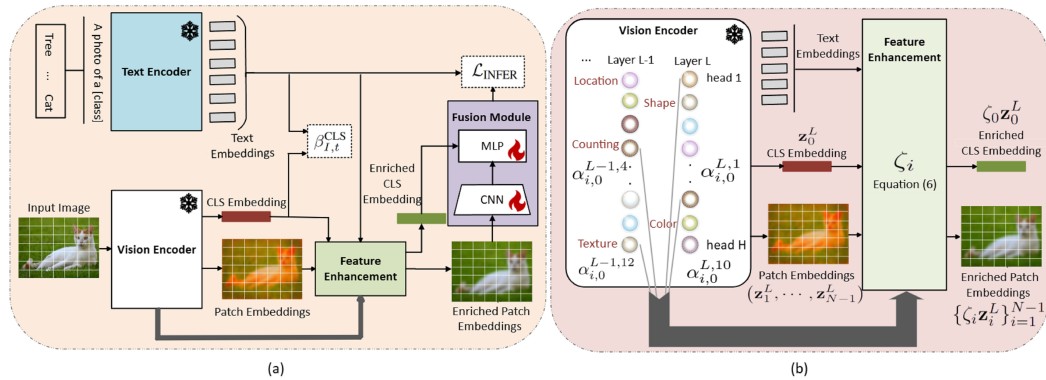

Figure 2: (a) INFER extends CLIP by leveraging both the CLS and patch embeddings. The vision encoder produces CLS and patch embeddings, which are first enriched by our feature enhancement mechanism. These enriched CLS and patch embeddings are then dynamically integrated through a lightweight fusion module, which consists of CNN-MLP networks, while keeping the pretrained CLIP encoders frozen. (b) Illustration of the feature enhancement mechanism. Attention weights across layers and heads capture diverse semantic features. For each embedding, coefficients $\zeta_i$ are computed by selecting the most informative heads, leading to enriched CLS and patch embeddings that retain fine-grained spatial and contextual cues.

embeddings can improve the model's discriminative capability, and (iii) Different attention heads contribute unequally to the final classification decision. INFER has three key contributions:

- **Dynamic Integration Mechanism:** We design a method that intelligently integrates the CLS embedding and other patch embeddings, allowing the model to leverage both global context and fine-grained spatial information based on the specific characteristics of each image.

- **Attention-based Feature Enhancement Mechanism:** To enhance the CLS and patch embeddings, we introduce a systematic method for identifying and selecting the most discriminative attention heads across multiple transformer layers, enabling the construction of more meaningful enhanced feature representations that align with target semantics.

- **Comprehensive Evaluation:** We provide extensive numerical results, which show that INFER sets new SOTA performance across diverse datasets, including fine-grained tasks.

## 2 RELATED WORK

### 2.1 ATTENTION HEAD ANALYSIS

The attention mechanism has become fundamental to modern vision architectures, including ViT, demonstrating that transformer architectures can achieve competitive performance compared to convolutional networks Dosovitskiy et al. (2020). Some attention heads focus on local texture patterns, while others capture global shape information or semantic relationships between objects. The method proposed in Clark et al. (2019) demonstrated that many attention heads in Bidirectional Encoder Representations from Transformers (BERT) can be removed without significant performance degradation, suggesting that not all heads contribute equally to model performance. Similar findings are reported for ViT, where selective modification of attention heads can improve both efficiency and accuracy Michel et al. (2019). TextSpan Gandelsman et al. (2023) provided important insights into the decomposition of CLIP's final image representation into contributions from individual image tokens. Their work demonstrated that patch-level information can be effectively utilized for interpretability and localization tasks, suggesting that similar principles could be applied to enhance classification performance. The PaCa-ViT Chefer et al. (2021) particularly emphasizes the importance of meaningful cluster assignments and attention patterns, showing that learned clusters can provide semantically meaningful visual tokens beyond simple patches. CLIP-PGS Pei et al. (2025) further supports this with their attention-based feature enhancement mechanism, where they identify the most informative attention heads based on alignment with class text embeddings.

## 2.2 PATCH-INCLUSIVE REPRESENTATION LEARNING FOR IMAGE SEGMENTATION

Recent studies in image segmentation tasks have provided valuable insights into leveraging CLIP's patch-level representations for patch-based representation learning. MaskCLIP Dong et al. (2023) pioneered the extraction of dense labels from CLIP by directly utilizing patch embeddings, demonstrating that CLIP's internal representations contain rich spatial information that can be effectively used for segmentation. SegViT Zhang et al. (2022) proposed an attention-to-mask module that directly utilizes spatial attention patterns for segmentation. Their approach demonstrated that attention mechanisms can be used not only for feature extraction but also for generating spatial masks, providing a natural bridge between attention patterns and spatial understanding. CLIP-PGS Pei et al. (2025) introduces a patch generation-to-selection strategy that enhances CLIP's training efficiency while preserving critical semantic content. This work demonstrates that careful patch selection can improve both efficiency and semantic integrity.

## 2.3 ADAPTATION AND FEW-SHOT FINE-TUNING STRATEGIES

Few-shot fine-tuning has heavily been explored for CLIP adaptation, with methods such as CLIP-Adapter Gao et al. (2021) introducing lightweight residual-style adapters that can be trained with limited data. Tip-Adapter Zhang et al. (2021) further advanced this direction by proposing a training-free adaptation approach that constructs adapters from cached few-shot training features. MMA Yang et al. (2024) introduced lightweight multi-modal adapters into both vision and text encoders to better align and adapt representations with minimal parameter cost. LP++ Huang et al. (2024) proposes a strong linear probing baseline that blends image and text prototypes through convex optimization for few-shot tasks. CLIP-LoRA Zanella & Ben Ayed (2024) applies low-rank parameterization to tune VLMs efficiently, reducing trainable parameters while preserving adaptation capacity. Most recently, 2SFS Farina et al. (2025) splits adaptation into two stages by first fine-tuning LayerNorms for general features, then training a classifier for specialization.

Prompt-based adaptation has gained significant attention as an alternative to parameter updates. CoOp Zhou et al. (2022c) introduced learnable prompts that can be optimized to adapt CLIP's text encoder to specific tasks, while CoCoOp Zhou et al. (2022b) extended this approach to conditional prompts that vary based on input images. to reduce forgetting. KgCoOp Yao et al. (2023) enhances prompt tuning by incorporating external knowledge into context vectors, resulting in more semantically meaningful and robust prompts.

## 3 EMBEDDING **IN**TEGRATION WITH **FE**ATURE **R**EFINEMENT (INFER)

We propose INFER based on the following insights: (i) Patch embeddings retain valuable spatial information that can complement global representations captured by CLS embedding, (ii) Different heads at different layers capture different information that can be selectively used to reweight the CLS and patch embeddings to improve performance, and (iii) Attention patterns encode diverse and meaningful spatial relationships useful for prediction tasks. Building on these insights, the proposed INFER is an approach that effectively utilizes patch embeddings as well as CLS embedding for fine-grained classification. INFER first enriches the patch and CLS embeddings by our proposed feature enhancement mechanism with effective head selection. Then INFER dynamically integrates the enriched CLS and patch embeddings through a lightweight fusion module. INFER offers a new solution for few-shot learning in VLMs.

### 3.1 STANDARD APPROACH

Consider a VLM, such as CLIP, which is composed of a vision encoder processing images $I$ and a text encoder processing text $t$. For a ViT-based vision encoder, the output of layer $l = 1, \cdots, L$ is denoted by $Z^l = \left[ \mathbf{z}_0^l, \mathbf{z}_1^l, \cdots, \mathbf{z}_{N-1}^l \right] \in \mathbb{R}^{d \times N}$, where $\mathbf{z}_0^l \in \mathbb{R}^d$ is the CLS embedding, and $\mathbf{z}_i^l \in \mathbb{R}^d, i = 1, \cdots, N-1$ represent the $(N-1)$ patch embeddings in layer $l$. In the standard setup, the CLS embedding $\mathbf{z}_0^L$ at the last layer of ViT is projected, using projection matrix $P \in \mathbb{R}^{d' \times d}$, onto the joint image-text embedding space, yielding the image representation $M_{\text{img}}^{\text{CLS}}(I)$ as follows:

$$M_{\text{img}}^{\text{CLS}}(I) = P\mathbf{z}_0^L \in \mathbb{R}^{d'}. \tag{1}$$

Given the text representation, $M_{\text{txt}}(t)$, for input text $t$, their cosine similarity is determined by

$$\beta_{I,t}^{\text{CLS}} = \cos\left(M_{\text{img}}^{\text{CLS}}(I), M_{\text{txt}}(t)\right) \in \mathbb{R}, \tag{2}$$

and this cosine similarity is used for model training and inference.

A drawback of this standard approach is that the image representation relies exclusively on the CLS embedding $\mathbf{z}_0^L$, without utilizing the remaining patch embeddings $\mathbf{z}_i^L, i = 1, \ldots, N-1$, which preserve fine-grained spatial cues critical for detailed image understanding. To address the limitation of the standard approach, we propose INFER by intelligently leveraging the information from all patch embeddings as well as the CLS embedding, thereby capturing fine-grained spatial cues.

## 3.2 INFER: FEW-SHOT EARNING FRAMEWORK

INFER introduces a few-shot learning mechanism that intelligently combines the enhanced CLS and patch embeddings, enabling dynamic alignment of global and local visual representations. Figure 2(a) provides an overview of INFER, while Figure 2(b) illustrates the proposed feature enhancement mechanism in detail.

The goal of INFER is to leverage both the patch embeddings $\{\mathbf{z}_i^L\}_{i=1}^{N-1}$ and the CLS embedding $\mathbf{z}_0^L$ from the final layer to construct the image representation, $M_{\text{img}}^{\text{INFER}}(I)$, which can be expressed as

$$M_{\text{img}}^{\text{INFER}}(I) = P \cdot g\left(\mathbf{z}_0^L, \mathbf{z}_1^L, \cdots, \mathbf{z}_{N-1}^L\right) \in \mathbb{R}^{d'}, \tag{3}$$

where $g : \mathbb{R}^{d \times N} \to \mathbb{R}^d$ denotes a function that merges all embeddings to a vector of dimension $d$. In our approach, before actual combining, each embedding $\mathbf{z}_i^L$ is first scaled (or enhanced) by a coefficient $\zeta_i$ for $i = 0, \cdots, N-1$, where $\zeta_i$ selectively enhances $\mathbf{z}_i^L$ by leveraging attention weights (from the CLS token to the $i$-th patch token) of select heads of ViT. This procedure captures the patch-level relevance to the CLS embedding. The enhanced embeddings $\{\zeta_i \mathbf{z}_i^L\}_{i=0}^{N-1}$ are combined by a small fusion module composed of a 3-layer CNN, $f_{\text{CNN}}$, and a single MLP layer, $f_{\text{MLP}}$. Specifically, $f_{\text{CNN}}$ is first used to combine the enhanced patch embeddings $\{\zeta_i \mathbf{z}_i^L\}_{i=1}^{N-1}$, and then $f_{\text{MLP}}$ is used to fuse the CNN output with the enhanced CLS embedding $\zeta_0 \mathbf{z}_0^L$ as follows:

$$M_{\text{img}}^{\text{INFER}}(I) = P \cdot f_{\text{MLP}}\left(\zeta_0 \mathbf{z}_0^L, f_{\text{CNN}}\left(\zeta_1 \mathbf{z}_1^L, \cdots, \zeta_{N-1} \mathbf{z}_{N-1}^L\right)\right) \in \mathbb{R}^{d'}. \tag{4}$$

The reason for using a CNN to first combine $\{\zeta_i \mathbf{z}_i^L\}_{i=1}^{N-1}$ is that patch embeddings inherently retain 2D spatial structure, since each embedding corresponds to a spatial patch of the input image. Rather than flattening or averaging these spatially distributed features (which discard spatial relations), CNN captures localized patterns and relationships across neighbouring enhanced patches.

Few-shot learning of INFER consists of two stages: In Stage 1, $\zeta_i$ are determined leveraging the attention weights and, in Stage 2, $f_{\text{CNN}}$ and $f_{\text{MLP}}$ are trained. Each stage is detailed in the following.

**(Stage 1) Feature Enhancement by** $\zeta_i$: To enrich both the CLS embedding and the patch embeddings, INFER leverages the attention weights from the CLS token to the $i$-th patch token, which quantify how much information each patch contributes to the global representation. This provides three key benefits: (i) CLS-to-patch attention highlights the most informative regions, implicitly capturing object boundaries or salient semantic parts, (ii) patch embeddings can be reweighted to incorporate their relative importance in the global context, leading to more discriminative and contextually meaningful features, and (iii) patches with high CLS attention often correspond to semantically critical elements, such as distinctive object parts or complementary components of a scene. In this way, INFER harnesses CLS-guided attention to selectively enhance feature representations, yielding enriched embeddings that better capture both global and local visual cues.

To quantify how strongly CLS token attends to the $i$-th patch token at each head at each layer, we consider attention weight $\alpha_{i,0}^{l,h} = \text{softmax}_i\left(\frac{1}{\sqrt{d_H}}(\mathbf{k}_i^{l,h})^T(\mathbf{q}_0^{l,h})\right)$, where $\mathbf{q}_0^{l,h} \in \mathbb{R}^{d_H}$ is the query vector of CLS token, $\mathbf{k}_i^{l,h} \in \mathbb{R}^{d_H}$ is the key vector of the $i$-th token, $d_H = \frac{d}{H}$ is the per-head dimensionality, and $H$ is the number of heads per layer. An approach to determine $\zeta_i$ is to take into account all attention weights $\alpha_{i,0}^{l,h} \in \mathbb{R}$ of all heads and all layers:

$$\zeta_i = \frac{1}{L \cdot H} \sum_{l=1}^{L} \sum_{h=1}^{H} \alpha_{i,0}^{l,h}, \quad i = 0, \cdots, N-1. \tag{5}$$

In INFER, this strategy of capturing CLS-to-patch attention is further improved in two key ways. First, in the ViT, deeper layers usually capture more semantic and global information, as the CLS attention scores in later layers better reflect the model's final "understanding" and representation of the image Gandelsman et al. (2023). Based on this inspiration, only the last $\hat{L}$ layers (e.g., four layers) are leveraged rather than all $L$ layers. Second, we note that not all attention heads contribute equally: different heads specialize in capturing different types of information and some heads may contain less meaningful information or just noise for certain classes. Therefore, treating all heads equally can attenuate the signal from the most informative ones. In light of this, we select a set of $J$ most informative heads at each layer for the $i$-th embedding, denoted by $\mathcal{H}_i^l$, thereby focusing on the most relevant attention patterns for each embedding. Together, $\zeta_i$ is refined as:

$$\zeta_i = \frac{1}{\hat{L} \cdot J} \sum_{l=L-\hat{L}+1}^{L} \sum_{h \in \mathcal{H}_i^l} \alpha_{i,0}^{l,h}, \;\; i = 0, \cdots, N-1. \tag{6}$$

To construct $\mathcal{H}_i^l$, we apply the few-shot samples to the frozen ViT and extract the attention weights $\alpha_{i,0}^{l,h}$, the patch embeddings $\mathbf{z}_i^L$, and the text embedding $M_{\text{txt}}(t)$. We then project the weighted patch embeddings $\alpha_{i,0}^{l,h}\mathbf{z}_i^L$ onto the joint embedding space to obtain $\mathbf{e}_i^{l,h} = P\alpha_{i,0}^{l,h}\mathbf{z}_i^L \in \mathbb{R}^{d'}$, and compute its cosine similarity with the text embedding $M_{\text{txt}}(t)$:

$$\phi_i^{l,h} = \cos\left(\mathbf{e}_i^{l,h}, M_{\text{txt}}(t)\right) \in \mathbb{R}, \;\; i = 0, \cdots, N-1; h = 1, \cdots, H; l = L-\hat{L}+1, \cdots, L. \tag{7}$$

We now construct the set $\mathcal{H}_i^l$ of head indices by selecting, for each embedding for each layer, the $J$ heads with the largest $\phi_i^{l,h}$ values as follows:

$$\mathcal{H}_i^l = \text{top}_J\left(\{\phi_i^{l,h}\}_{h=1}^H\right), \;\; i = 0, \cdots, N-1; l = L-\hat{L}+1, \cdots, L, \tag{8}$$

where $\text{top}_J(\cdot)$ denotes the function that returns the set of indices corresponding to the largest $J$ values from the input set.

In INFER, feature enhancement is performed per class to capture class-specific discriminative features by selecting informative heads and reweighting patch and CLS embeddings. Assuming that there are $C$ classes, few-shot samples are given for each class $c \in \{1, \cdots, C\}$. The attention weights, now denoted by $\alpha_{i,0}^{l,h,c}$, from CLS token to the $i$-th token of head $h$ at layer $l$ for class $c$, are determined and stored. This process yields $\mathbf{e}_i^{l,h,c}$, $\phi_i^{l,h,c}$, $\mathcal{H}_i^{l,c}$, and $\zeta_i^c$. For a given image $I$, INFER then generates $C$ class-specific image embeddings $M_{\text{img}}^{\text{INFER},c}(I), c = 1, \cdots, C$, constructed from the corresponding stored values of $\zeta_i^c$.

**(Stage 2) Integration of Enhanced Features by $f_{\text{CNN}}$ and $f_{\text{MLP}}$:** INFER integrates the enriched CLS and patch embeddings obtained from Stage 1 to capture both global and local cues. To achieve this, a lightweight fusion module is introduced (Figure 2(a), Equation 4). Specifically, $f_{\text{CNN}}$ is applied to combine the patch embeddings to preserve their spatial structure and local interactions, while $f_{\text{MLP}}$ combines the resulting representation with the CLS embedding. This design ensures that fine-grained local information is aligned with global semantics, yielding more discriminative representations for classification.

Given few-shot samples, the cosine similarity between $M_{\text{img}}^{\text{INFER},c}(I)$ and $M_{\text{txt}}(t)$ is determined as:

$$\beta_{I,t}^{\text{INFER},c} = \cos\left(M_{\text{img}}^{\text{INFER},c}(I), M_{\text{txt}}(t)\right) \in \mathbb{R}. \tag{9}$$

Following the standard CLIP loss formulation, the image-to-text loss $\mathcal{L}_{\text{image}}$ and text-to-image loss $\mathcal{L}_{\text{text}}$ are computed, but with $\sum_{c=1}^{C} \beta_{I,t}^{\text{INFER},c}$ replacing $\beta_{I,t}^{\text{CLS}}$. The final training loss is $\mathcal{L}_{\text{INFER}} = \mathcal{L}_{\text{image}} + \mathcal{L}_{\text{text}}$. Using $\mathcal{L}_{\text{INFER}}$, we first train $f_{\text{CNN}}$ with the few-shot samples. With $f_{\text{CNN}}$ frozen, we then train $f_{\text{MLP}}$ on the same samples using $\mathcal{L}_{\text{INFER}}$ again.

### 3.3 INFER: INFERENCE FRAMEWORK

Given an unknown image $I$ and a set of $C$ candidate classes, INFER produces $C$ class-specific image embeddings, $M_{\text{img}}^{\text{INFER},c}(I), c = 1, \cdots, C$, constructed from the stored values of $\zeta_i^c$. To

evaluate alignment between these embeddings and the candidate text embeddings, we compute the cosine similarity as follows:

$$\beta_{I,t}^{\text{INFER},c,c'} = \cos\left(M_{\text{img}}^{\text{INFER},c}(I), M_{\text{txt}}(t_{c'})\right) \in \mathbb{R}, \quad c \in \{1, \cdots, C\}, \ c' \in \{1, \cdots, C\}, \quad (10)$$

where $t_{c'}$ denotes the text corresponding to the $c'$-th class. For prediction, INFER aggregates the similarities across all class-specific image embeddings and selects the class with the highest alignment with the text embedding:

$$c^{\text{opt}} = \arg\max_{c' \in \{1, \cdots, C\}} \sum_{c=1}^{C} \beta_{I,t}^{\text{INFER},c,c'}. \quad (11)$$

This inference rule effectively chooses the class whose text embedding achieves the strongest overall alignment with all class-specific views of the image.

## 4 EXPERIMENTS

### 4.1 IMPLEMENTATION DETAILS AND DATASETS

For experiments, we adopt both CLIP ViT-B/32 (with 12 layers with 12 attention heads per layer) and CLIP ViT-L/14 (with 24 layers with 16 attention heads per layer). For few-shot fine-tuning, we used 16 samples per class for each dataset. The batch size is 32 with a total iteration of 100. We used an Adam optimizer with a learning rate of 0.001 and the cross-entropy loss. INFER leverages the best $J = 4$ attention heads per layer from the final $\hat{L} = 4$ layers for feature enhancement. The rationale for this selection is provided in Section 4.3.3. Model performance is evaluated using overall accuracy as the metric.

A broad range of datasets are considered, including ImageNet (Deng et al., 2009), with 1,000 classes, OxfordPet (Parkhi et al., 2012), with 37 breeds of cats and dogs, Cars (Krause et al., 2013), with 196 car models categorized by make, model, and year, Flowers102 (Nilsback & Zisserman, 2008) containing 102 flower categories, Food101 (Bossard et al., 2014), with 101 food categories, Caltech101 (Fei-Fei et al., 2004) which contains 101 object categories plus one background/clutter category, SUN397 (Xiao et al., 2010),which includes 397 scene categories, UCF101 Soomro (2012), with 101 human action categories, EuroSat Helber et al. (2019) with 10 land-use and land-cover categories, and FGVCAircraft (Maji et al., 2013) which contains 100 aircraft variants, emphasizing for fine-grained recognition between visually similar categories.

### 4.2 RESULTS

Table 1 reports the results of INFER obtained from our own simulations, while the remaining results are reproduced from Table 2 of 2SFS Farina et al. (2025). Although 2SFS is the current SOTA, INFER achieves superior performance over it and all other baselines, establishing new SOTA.The results highlight INFER's ability to capture fine-grained spatial cues that are often missed when relying solely on the CLS embedding. On scene-level and generic object recognition datasets (e.g., SUN, EuroSAT, Flowers, Caltech), INFER surpasses the strong baselines, showing that the method generalizes beyond fine-grained classification. On fine-grained recognition datasets (e.g., CARS, FGVC, DTD, UCF), INFER also consistently achieves the best results, outperforming SOTA baselines. Its strength lies in an attention-guided feature enhancement and the dynamic integration of the enhanced CLS and patch embeddings, enabling more expressive and robust multimodal representations.

### 4.3 ABLATION STUDIES

#### 4.3.1 EFFECTIVENESS OF THE PROPOSED DYNAMIC INTEGRATION MECHANISM

Figure 1(a) presents heatmaps of the standard CLS embeddings and the INFER embeddings, showing that INFER effectively leverages patch information to produce richer representations. In Figures 3(a)–(d), the confusion matrix and the t-SNE visualization at the last layer of standard CLIP and INFER are presented for the MNIST dataset. It is clear that standard CLIP is heavily biased towards some classes, while INFER can effectively diagonalize the confusion matrix.

| Backbone | Method | ImageNet | SUN | FGVC | EuroSAT | CARS | FOOD | PETS | Flowers | Caltech | DTD | UCF | Mean |
|---|---|---|---|---|---|---|---|---|---|---|---|---|---|
| | CLIP Zero-Shot | 61.9 | 62.0 | 19.3 | 45.1 | 60.4 | 80.5 | 87.5 | 67.0 | 91.1 | 42.6 | 62.2 | 61.8 |
| | CoOp | 66.8 | 69.2 | 30.8 | 73.4 | 64.6 | 81.9 | 91.0 | 82.5 | 94.3 | 59.7 | 75.3 | 71.8 |
| | CoCoOp | 66.1 | 69.8 | 34.5 | 74.0 | 64.0 | 81.7 | 91.4 | 82.3 | 94.1 | 59.0 | 75.5 | 72.0 |
| | TIP-Adapter-F | 64.0 | 71.4 | 29.8 | 71.7 | 68.0 | 81.9 | 90.1 | 88.7 | 94.8 | 58.1 | 76.5 | 72.3 |
| | CLIP-Adapter | 64.7 | 71.4 | 30.7 | 71.8 | 68.9 | 81.7 | 90.1 | 88.7 | 94.8 | 58.1 | 76.5 | 72.5 |
| | KgCoOp | 65.4 | 71.3 | 32.0 | 70.1 | 77.3 | 81.7 | 90.8 | 86.1 | 94.4 | 65.1 | 77.5 | 73.8 |
| ViT-B/32 | MaPLe | 66.7 | 72.2 | 36.0 | 81.3 | 66.9 | _82.1_ | _91.9_ | 95.0 | 95.1 | 65.8 | 77.3 | 75.1 |
| | ProGrad | 68.1 | 73.2 | 38.0 | 82.0 | 66.7 | 80.2 | 91.1 | _96.3_ | 95.5 | 65.6 | 77.2 | 75.5 |
| | CLIP-LoRA | 68.4 | 74.0 | 37.2 | _92.8_ | 75.2 | 81.7 | 90.5 | 96.0 | 95.2 | 68.2 | 80.2 | 78.1 |
| | MMA | _68.8_ | 74.5 | _40.2_ | 90.1 | 73.5 | 81.4 | 91.5 | _96.3_ | 95.6 | 70.9 | 81.7 | 78.6 |
| | NormFit | 65.6 | 72.5 | 35.2 | 75.1 | 74.4 | 80.9 | 89.8 | 89.3 | 95.5 | 69.4 | 80.5 | 76.7 |
| | 2SFS | 68.4 | _74.8_ | 40.2 | 92.1 | _80.2_ | 80.8 | 90.3 | _96.3_ | _95.8_ | _70.4_ | _82.3_ | _79.2_ |
| | INFER (ours) | **70.2** (±0.12) | **75.2** (±0.18) | **40.6** (±0.08) | **94.2** (±0.13) | **82.7** (±0.10) | **82.8** (±0.05) | **92.6** (±0.06) | **96.7** (±0.15) | **96.3** (±0.17) | **71.2** (±0.09) | **83.6** (±0.10) | **80.5** (±0.11) |
| | CLIP Zero-Shot | 72.9 | 67.6 | 32.6 | 58.0 | 76.8 | 91.0 | 93.6 | 79.4 | 94.9 | 54.2 | 72.2 | 72.1 |
| | CoOp | 78.0 | 78.2 | 52.0 | 89.0 | 89.0 | _93.6_ | 99.1 | 97.5 | 97.4 | 77.3 | 88.4 | 85.4 |
| | CoCoOp | 77.3 | 78.6 | 54.4 | 89.3 | 89.2 | _93.6_ | 99.1 | 97.4 | 97.3 | 74.0 | 87.4 | 85.3 |
| | TIP-Adapter-F | 77.3 | 79.3 | 52.9 | 89.5 | 86.1 | 91.6 | 94.6 | 97.3 | _97.5_ | 74.0 | 87.8 | 84.4 |
| | CLIP-Adapter | 78.0 | 79.3 | 53.0 | 89.7 | 86.1 | 91.6 | 94.6 | 97.3 | _97.5_ | 74.0 | 87.8 | 84.5 |
| | KgCoOp | 78.3 | 79.6 | 53.6 | 89.0 | 88.0 | _93.6_ | _98.9_ | 97.4 | 97.4 | 75.0 | 87.3 | 85.3 |
| ViT-L/14 | MaPLe | 78.4 | 79.7 | 59.0 | 90.3 | 88.0 | 92.9 | 98.8 | 97.4 | _97.5_ | 75.4 | 87.8 | 85.9 |
| | ProGrad | 78.3 | 79.9 | 58.7 | 89.7 | 88.2 | 92.9 | _98.9_ | 97.5 | 97.4 | 75.5 | 87.6 | 85.9 |
| | CLIP-LoRA | _79.9_ | 79.9 | 58.8 | 92.8 | 89.2 | 93.2 | _98.9_ | 97.5 | _97.5_ | 75.8 | 88.0 | 86.5 |
| | MMA | 79.0 | _80.4_ | _64.1_ | _92.9_ | 89.0 | 92.9 | _98.9_ | 97.5 | _97.5_ | 75.8 | 88.0 | 86.9 |
| | NormFit | 78.2 | 78.8 | 56.1 | 90.2 | 88.8 | 91.6 | 97.9 | 97.5 | 97.4 | 74.5 | 87.9 | 85.4 |
| | 2SFS | 79.4 | 80.3 | _64.1_ | _92.9_ | _90.3_ | 91.1 | 95.5 | _99.1_ | _97.5_ | _78.0_ | _89.5_ | _87.1_ |
| | INFER (ours) | **80.9** (±0.05) | **81.5** (±0.03) | **65.2** (±0.02) | **95.2** (±0.03) | **91.7** (±0.04) | **94.0** (±0.03) | **99.6** (±0.04) | **99.6** (±0.02) | **98.0** (±0.02) | **80.1** (±0.03) | **91.0** (±0.05) | **88.8** (±0.03) |

Table 1: Classification accuracy across diverse datasets with ViT-B/32 and ViT-L/14 backbones.

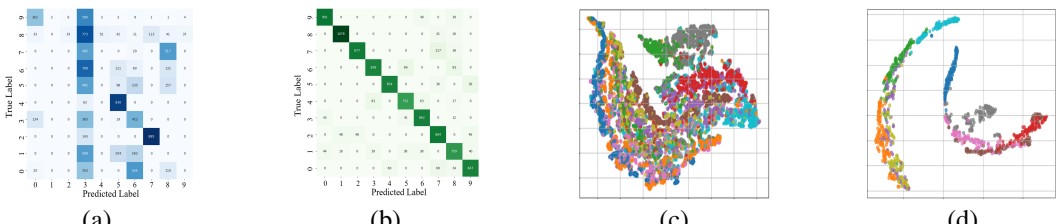

| (a) | (b) | (c) | (d) |

Figure 3: Confusion matrix of (a) standard CLIP, and (b) INFER. t-SNE visualization at the outputs of (c) standard CLIP, and (d) INFER.

### 4.3.2 CLASS-SPECIFIC ATTENTION HEAD SELECTION

The proposed feature enhancement mechanism dynamically selects optimal attention heads, and reweights the CLS and patch embedding such that the most discriminative attention patterns are preserved. Figure 4 shows the distribution of selected attention heads for 10 classes of CIFAR-10. Each subplot corresponds to a class, with the $x$-axis denoting the head index (1–12) and the $y$-axis representing the number of times each head is selected. The results indicate that different classes emphasize distinct subsets of heads, suggesting that INFER tailors its head selection to class-specific visual cues rather than relying on a fixed universal pattern. The variation across classes highlights both shared and specialized semantic structures captured by CLIP's attention heads. For instance, animal categories (e.g., cat, dog, frog, horse) frequently exploit a broader range of heads, reflecting the need to capture fine-grained textures and shapes, while vehicle categories (e.g., airplane, car, truck) concentrate more heavily on a few dominant heads that encode geometric and structural features. This demonstrates that INFER not only identifies universally informative heads but also dynamically exploits discriminative heads that align with semantic distinctions between object categories, thereby enhancing the robustness of the learned representations.

### 4.3.3 ABLATION ON $J$ AND $\hat{L}$

Figure 5 explores the effect of the hyperparameters $J$ and $\hat{L}$ on INFER. In Figure 5(a), accuracy increases with $J$ up to 4 but declines when more heads are used, suggesting that too many attention

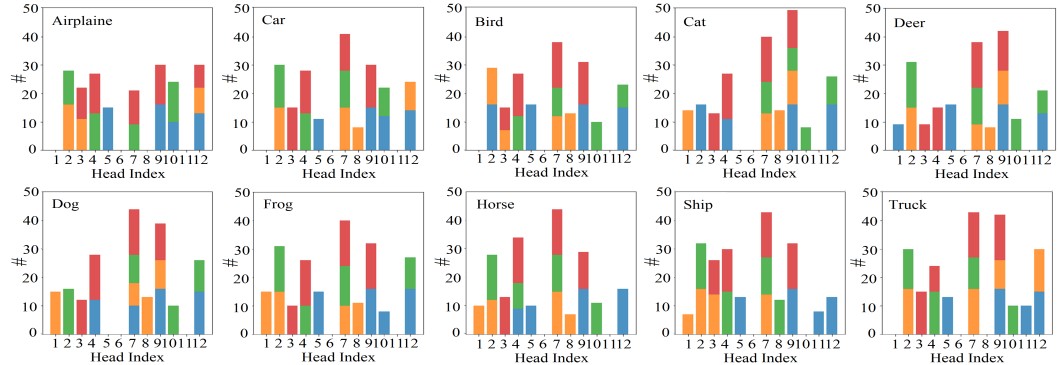

Figure 4: Head selection across ten CIFAR-10 classes in INFER with ViT-B/32. Each subfigure corresponds to one class. The $x$-axis denotes the head index. The $y$-axis shows the number of times each head is selected (and included $\mathcal{H}_i^{l,c}$) for the last four layers when the total number of trials is 16 for each class. For each head, the bars in blue, orange, green, and red represent Layers 12, 11, 10, and 9, respectively. For each class, a head may not be selected at all or may be selected at multiple layers. Results indicate that different heads of different layers specialize in different classes, and INFER dynamically exploits this diversity through its feature enhancement mechanism.

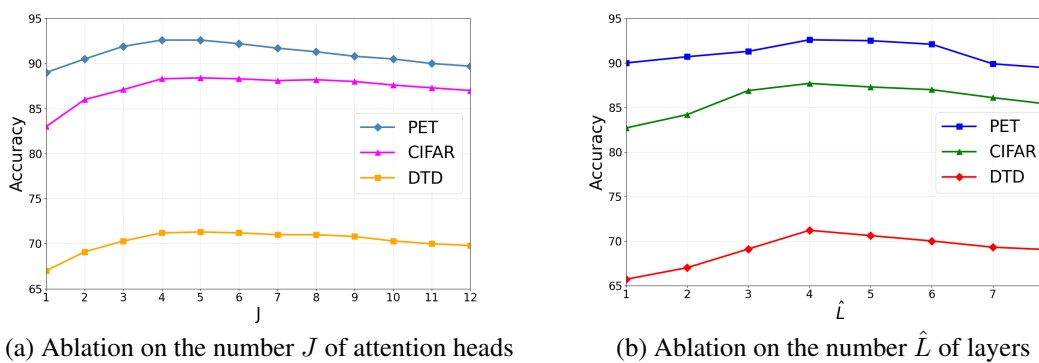

(a) Ablation on the number $J$ of attention heads  (b) Ablation on the number $\hat{L}$ of layers

Figure 5: Ablation studies of INFER. (a) Effect of the number $J$ of attention heads selected, showing that accuracy improves up to $J = 4$ and then declines. (b) Effect of the number $\hat{L}$ of layers leveraged, with peak accuracy observed at $\hat{L} = 4$.

heads introduce noise and reduce performance. In Figure 5(b), accuracy peaks at $\hat{L} = 4$, indicating that the last four layers provide the most meaningful cues, while additional layers contribute redundancy and lead to a slight drop. These results establish $J = 4$ and $\hat{L} = 4$ as effective choices, validating the design of INFER.

## 5 CONCLUSION

We introduced INFER to improve CLIP's image representations by leveraging semantically informative attention heads to selectively enhance both CLS and patch embeddings. By operating entirely within CLIP's frozen backbone, INFER avoids architectural modifications and preserves the model's generalization ability. Our method not only improves the performance on few-shot classification benchmarks but also provides valuable interpretation into CLIP's internal attention dynamics. The simplicity and modularity of INFER make it applicable to a wide range of downstream tasks and foundation models. Future work may explore the extension of INFER to multi-modal retrieval or segmentation scenarios.

## REPRODUCIBILITY STATEMENT

We have made efforts to ensure the reproducibility of our work. Detailed descriptions of the INFER architecture, training setup, and evaluation protocols are provided in Sections 3 and 4. Additional ablation studies and visualizations are provided to further clarify design choices. The datasets used (ImageNet, SUN, FGVC-Aircraft, EuroSAT, CARS, Food, Pets, Flowers, Caltech, DTD, and UCF) are publicly available. For transparency, we release **anonymized complete source code and instructions** as part of the supplementary material, enabling independent verification of all experimental results.

## USE OF LARGE LANGUAGE MODEL

We used ChatGPT solely to polish grammar and wording.

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
