# OpenReview forum: "INFER: Embedding Integration with Feature Refinement for Few-Shot Learning in VLMs"
_ICLR.cc/2026/Conference — Submitted to ICLR 2026_

### Official Review · Reviewer_Gyds · 2025-10-19

**Soundness:** 3
**Presentation:** 2
**Contribution:** 2
**Rating:** 4
**Confidence:** 5

**Summary:**

The paper presents a novel framework, INFER, that enhances frozen CLIP embeddings for few-shot image recognition. The authors introduce two main innovations: first, they select class-consistent attention heads from the final layers of the CLIP visual transformer to reweight both class token and patch representations; second, they fuse the enhanced features using a lightweight CNN and MLP module to produce a more discriminative image embedding aligned with textual class semantics.
Through this attention-guided reweighting and dual-path fusion, INFER effectively leverages the latent spatial details already present in CLIP without additional fine-tuning. Experiments across ten benchmarks show that INFER consistently outperforms strong baselines and recent state-of-the-art few-shot methods, especially in fine-grained classification settings.

**Strengths:**

1. This paper introduces a novel attention-head selection mechanism to leverage class-consistent visual cues without modifying the pretrained encoder.
2. The proposed attention-based feature enhancement and CNN–MLP dual fusion modules provide a clear and interpretable way to reweight and integrate spatial and semantic information, improving representation quality.
3. The method is computationally efficient and training-free, requiring no additional gradient updates while achieving substantial gains in accuracy.

**Weaknesses:**

1. The paper title is generic. Integration, refinement, and VLMs are all common prospectives of few-shot learning and vision-language models. They struggle to effectively convey the unique contributions or technical highlights of the work. A more specific and informative title would help better position the paper.
2. The motivation is limited. Many existing CLIP-based models not only depend on CLS token for classification, but also leverage remaining token sequence, including exploration about local patch enhancement. The innovation is not powerful compared to relevant works. A more clear and specific comparison with similar works on theory and experiments is important.
3. INFER generates a class-specific image embedding for every candidate class and aggregates similarities, computation and storage grow proportionally to C It would be important to compare inference/storage overhead with a few existing papers on large vocabularies (e.g., ImageNet-1k)
4. Semantic information with a simple template ("A photo of a [class]") is limited helpful on discriminative and class-specific tokens, deviating the attention choice from the actual content needed.  The potential benefits of enriching semantic information deserve further exploration.
5. The effectiveness of INFER on fine-grained scenarios should be further verified on more benchmarks, such as CUB, Cars, and Dogs.

**Questions:**

yes.

**Details Of Ethics Concerns:**

yes.

---

> ### Author Response · Authors · 2025-11-21
> **Responses to [Weakness 1 and Weakness 2-Part 1]. Thank you for the comments.**
>
> # [Response to Weakness 1]
>
> Thank you for the comment. We revise the title to better describe the technical contributions of our work as follows:
> * **Revised Title:** "INFER: Selective Attention-Head Fusion and Patch–Token Reintegration for Fine-Grained CLIP Adaptation"
> The new title explicitly reflects our two core innovations, selective attention-head fusion and patch–token reintegration. We retain the name of our method, INFER (INformative Feature REintegration), as it effectively identifies our approach.
>
> ---
>
> # [Response to Weakness 2]-Part 1
>
> Thank you for raising this point. In what follows, we clarify (1) the gap we target, (2) how INFER differs from existing token/patch–aware approaches, and (3) how this is reflected in theory and experiments.
>
> **1. Clarifying the gap and motivation**
>
> Our paper focuses specifically on **few-shot classification with a frozen CLIP**, and to the best of our knowledge, all existing works operate on the text side (prompt tuning) or entirely on the final CLS token as the image representation. **They do not** use intermediate patch tokens or attention-head structure during inference.
>
> There is a separate line of work that does use non‑CLS tokens and local patches, but exclusively for *segmentation or interpretability* (MaskCLIP, SegViT, CLIP‑PGS, TextSpan, etc.) rather than few-shot image classification. These works show that CLIP’s patch tokens and attention maps are rich, but they:
>
> * typically aggregate patches in a task-specific way (e.g., to form masks),
>
> * do not build class-specific enhanced image embeddings for classification,
>
> * do not provide a lightweight, frozen-backbone few-shot framework that systematically combines CLS and patch embeddings for recognition.
>
> Our motivation is precisely to bridge this gap: take the strong few-shot classification setting of CLIP, and systematically bring in the underused patch and attention-head structure, *without* modifying or fine-tuning the backbone.
>
> In the revised paper, we will clarify this motivation in Sections 1 and 2.
>
> **2. What is actually new in INFER (beyond using non‑CLS tokens)**
> We agree that "simply using patch tokens” is not a strong contribution by itself. INFER’s novelty is **not** in simply using of patches, **but in how** CLS and patch embeddings are refined and integrated:
>
> **2.1 Text-conditioned, class-specific attention-head selection**
>
> * For each layer and head, we project attention-weighted patch embeddings into the joint image–text space and measure their cosine similarity with the corresponding class text embedding.
>
> * For each patch and for each class, we then select the *top‑$J$ most discriminative heads* whose contributions are best aligned with the text (Eq. (8)), and use only these to form the enhancement coefficients $\\{\zeta_i\\}$ (Eq. (6)).
>
> This then yields *class-specific* $\\{ \zeta_i^c \\}$ that encode “how much this patch matters for this class, according to the most text-aligned heads in the last $\hat L$ layers.” This is different from:
>
> * uniform or heuristic head averaging,
>
> * using a fixed set of “good” heads for all classes, or
>
> * using attention maps only as visualization/masks without coupling them to text embeddings.
>
> **2.2 Two-stage, frozen-backbone feature refinement framework**
>
> INFER keeps the CLIP backbone *entirely frozen* and separates adaptation into:
>
> * Stage 1: Feature enhancement
>
>  * We run few-shot samples once to compute and store the class-specific $\\{ \zeta_i^c \\}$ and the selected head sets.
>
>
> * Stage 2: Dynamic fusion
>
>
>   * A small 3‑layer CNN, $f_{\text{CNN}}$, combines the enhanced patch embeddings (keeping the 2D structure), and an MLP, $f_{\text{MLP}}$, fuses this with the enhanced CLS token to form $M^{\text{INFER},c}_{\text{img}}(I)$ (Eqs. (4) and (9)).
>
>
> This design is different from existing adapter/prompt methods that (i) insert residual blocks inside the backbone, or (ii) only alter text prompts while reusing the vanilla CLS embedding.
>
>
>
> ***Due to the character limit of this rebuttal box, our response continues in Part 2 in the next rebuttal entry.***

---

> ### Author Response · Authors · 2025-11-21
> **Response to [Weakness 2-Part 2]. Thank you for the comment.**
>
> # [Response to Weakness 2]-Part 2 (Continued)
>
>
> **2.3 Multi-view, class-specific inference rule**
>
>
>
> At inference time, INFER does not use a single image embedding; instead, it constructs $C$ class-specific image embeddings $M^{\text{INFER},c}_{\text{img}}(I)$ and aggregates their similarities (Eqs. (10) and (11)). This “multi-view” decision rule leverages the fact that different classes emphasize different heads and patches, which we empirically confirm in Fig. 4.
>
>
>
> To the best of our knowledge, prior CLIP-based few-shot methods do **not** perform such class-specific image embedding construction combined with attention-head selection and text-guided weighting.
>
>
>
> In revision, we will make these differences more explicit in the theory section (Sec. 3) and in the related work discussion.
>
>
>
> **3. Empirical evidence**
>
>
>
> **3.1 Performance improvements over the SOTA baselines**
>
>
>
> On the extensive set of 11 datasets in Table 1, INFER consistently outperforms the strongest existing methods for both ViT‑B/32 and ViT‑L/14 backbones. These improvements are non-trivial in the highly competitive regime where gains over strong baselines are typically small.
>
>
>
> **3.2 Ablations isolating the contribution of head selection and layer choice**
>
>
>
> Fig. 5 shows that:
>
>
>
> * Accuracy increases when we move from using all heads to selecting only the top‑$J$ heads, peaking at $J = 4$, after which performance declines, indicating that our head selection is crucial and that “using more heads” is not always better.
>
> * Leveraging only the last $\hat L = 4$ layers outperforms using shallower or all layers, consistent with the intuition that deeper layers encode higher-level semantics that align better with text.
>
>
>
> **3.3 Qualitative improvements in spatial focus and representation structure**
>
>
>
> * Heatmaps in Fig. 1(a) show that INFER produces *more compact and semantically aligned spatial activations* than standard CLIP, confirming that our feature enhancement indeed focuses on discriminative regions.
>
> * Confusion matrices and t‑SNE plots in Fig. 3 demonstrate that INFER reduces bias and yields *better-separated class clusters*, especially on fine-grained tasks.
>
>
>
> In revision, we will report these findings in the experimental section.
>
>
>
> **4. In the revised paper:**
>
> We will expand the related work** (Sec. 2.1–2.3) to explicitly separate CLS-only few-shot adaptation methods, and patch/attention–based segmentation/interpretability methods,
>
> We also add a dedicated subsection** comparing INFER conceptually to patch-enhancement works such as MaskCLIP, SegViT, and CLIP‑PGS.

---

> > ### Author Response · Authors · 2025-11-21
> > **Response to [Weakness 3]. Thank you for the comment.**
> >
> > # [Response to Weakness 3]
> >
> > Thank you for this important comment. Below we clarify the inference complexity and the storage overhead.
> >
> >
> > **1. Inference Complexity**
> >
> > During few-shot training (before inference), we do:
> >
> > 0. Precompute per-class enhancement coefficients $\\{\zeta_i^c\\}_{i=0}^{N-1}$.
> >
> > In inference, we do:
> >
> > 1.	Run the CLIP ViT encoder **once** to obtain the final-layer CLS and patch embeddings $\\{ z_i^L\\}_{i=0}^{N-1}$.
> >
> > 2. Obtain class-specific enhanced features by simple multiplications of the coefficients: $\\{\zeta_i^c z_i^L\\}_{i=0}^{N-1}$.
> >
> > 3. Pass these through a very small fusion module (3-layer CNN + 1-layer MLP) to obtain $C$ class-specific image embeddings $M_{\text{img}}^{\text{INFER},c}(I)$.
> >
> > 4. Compute cosine similarities between the $C$ image embeddings and the $C$ precomputed text embeddings, as in Eqs. (10) and (11).
> >
> > In inference, therefore, the **dominant cost** of the ViT backbone is incurred **only once**, **exactly the same as** in standard CLIP, linear probes, and 2SFS. The additional class-dependent work is confined only to very lightweight modules and dot products. Below is more formal analysis of the inference complexity:
> >
> > Let
> >
> > - $F_{\text{ViT}}$: FLOPs for one CLIP ViT forward pass
> >
> > - $d'$: joint embedding dimension (e.g., 512)
> >
> > - $F_{\text{fusion}}$: FLOPs of the CNN+MLP fusion module per class.
> >
> > Then, the inference complexity is:
> >
> > - Linear probe and 2SFS: $F_{\text{ViT}} + O(C d')$
> >
> > - INFER: $F_{\text{ViT}} + F_{\text{fusion}} +  O(C^2 d')$
> >
> >
> >
> > In INFER, the second term $F_{\text{fusion}}$ (due to the fusion) adds negligible complexity compared to other terms. While the third term $O(C^2 d')$ of INFER is not negligible, it is numerically far smaller than $F_{\text{ViT}}$.
> >
> > Even for ImageNet-1k (where $C$ is as large as 1,000), the third term $O(C^2 d')≈1.05$ GFLOPs for $d'=512$ or $≈1.53$ GFLOPs  for $d'=768$. Meanwhile, $F_{\text{ViT}}≈81.1$ GFLOPs for CLIP ViT-L/14,  $≈33.7$ GFLOPs for ViT-B/16, and $≈8.7$ GFLOPs for ViT-B/32, which correspond to additional 1.9%, 3.0%, and 11.7% overheads, respectively. For fine-grained dataset such as CARs (with $C$=196), they additional overheads reduce to 0.07%, 0.12%, and 0.45%, respectively.
> >
> > In the revised paper, we will clarify that INFER’s inference complexity is only slightly higher than linear probe or 2SFS.
> >
> >
> > **2. Storage Overhead**
> >
> >
> > INFER introduces additional per-class quantities with the storage growth is linear in $C$, just as in standard linear probes and 2SFS. Below is more formal analysis.
> >
> > - INFER stores per-class enhancement coefficients $\\{\zeta_i^c\\}_{i=0}^{N-1}$, i.e., a matrix of size $C \times N$, where $N$ is the number of tokens (patches + CLS). For CLIP ViT-B/32 with 224×224 inputs, $N = 50$; for ViT-L/14, $N = 257$.
> >
> > - In contrast, linear probes and 2SFS store a classifier weight matrix of size $C \times d'$, where $d'$ is the joint embedding dimension (e.g., $d' = 512$).
> >
> > Since typically $N \ll d'$, the per-class parameters of INFER $(C \times N)$ are actually smaller than those of a standard linear classifier $(C \times d')$. For ImageNet-1k, this corresponds to storing on the order of $10^5–10^6$ extra scalars, i.e., well under a couple of megabytes in fp16/fp32, which is negligible compared to the hundreds of megabytes of the frozen CLIP backbone.
> >
> > Overall, INFER’s per-class parameters scale linearly with $C$ but are not worse than a linear classifier and in fact smaller: $(C \times N)$ vs $(C \times d')$.
> >
> > In the revised paper, we will add a comparison table (INFER vs linear probe vs 2SFS)  to explicitly quantify this storage overhead.

---

> ### Author Response · Authors · 2025-11-21
> **Responses to [Weaknesses 4 and 5]. Thank you for the comments.**
>
> # [Response to Weakness 4]
>
>
>
> Thank you for this insightful comment. We agree that, in general, the semantic richness of text prompts can influence how image features are aligned and which attention heads are deemed informative. To directly evaluate the reviewer’s suggestion, we conducted new experiments on CIFAR-10 using INFER with CLIP ViT-B/32 in the 16-shot setting under two prompting regimes:
>
>
>
> **(i) Standard prompts (used in the main paper):** “a photo of a [class]”
>
>
>
> **(ii) LLM-enriched prompts (new):** For each class, we used an external LLM (ChatGPT) offline to generate several short, purely visual descriptions. We then constructed prompts of the form “a photo of a [class], [description]”, encoded each with the CLIP text encoder, and averaged the resulting embeddings per class to obtain a richer class prototype.
>
>
>
> Concretely, for CIFAR-10 we generated five descriptions for each class using ChatGPT 5.1. For example, for airplane:
>
>
>
> - "flying across blue sky with extended wings"
>
> - "passenger jet on runway with engines visible"
>
> - "small plane viewed from below against clouds"
>
> - "commercial airplane taking off above airport lights"
>
> - "sleek white aircraft parked near terminal"
>
>
>
> Replacing the standard template with these LLM-enriched prompts leads to a modest improvement, from 88.1% to 88.8% in the 16-shot setting with the ViT-B/32 backbone. This confirms that providing more detailed, class-specific visual semantics can further benefit INFER, as the reviewer suggested.
>
>
>
> However, the gain remains relatively modest. This indicates that INFER already uses the semantic information in the standard CLIP template effectively, and that its main improvements stem from visual feature refinement and fusion, rather than from sophisticated prompt engineering.
>
>
>
> It is important to note that incorporating LLM-generated prompts introduces additional methodological complexity. To keep evaluations **consistent and fair** with the baseline methods, we thus retain the standard CLIP template (“a photo of a [class]”) for all methods in the main paper.
>
>
>
> In the revised paper, following the reviewer’s comment, we will include the LLM-augmented prompt results as an ablation in the appendix, along with the full set of generated descriptions and an explicit note that these rely on an external LLM.
>
>
>
> ---
>
>
>
> # [Response to Weakness 5]
>
>
>
> We thank the reviewer for the suggestion to validate INFER on three fine-grained benchmarks: CUB, Stanford Dogs, and Stanford Cars.
>
>
>
>
> | Method | CUB   | Dogs  | Cars |
> |--------|-------|-------|-------|
> | 2SFS   | 74.49 | 75.70 | 80.2  |
> | INFER  | 76.12 | 78.30 | 82.7  |
>
>
>
> INFER consistently outperforms the current SOTA, 2SFS, on all three datasets.
>
> In the revised paper, we will include the results.

---

### Official Review · Reviewer_sEFX · 2025-10-22

**Soundness:** 2
**Presentation:** 2
**Contribution:** 2
**Rating:** 4
**Confidence:** 4

**Summary:**

This paper introduces INFER, a framework that enhances CLIP's few-shot learning capabilities by addressing its over-reliance on the global [CLS] embedding. INFER leverages rich spatial information from patch embeddings by proposing a two-stage approach: first, it selectively enhances both [CLS] and patch features using a class-specific attention head selection mechanism, which identifies heads that are most semantically aligned with the target class's text embedding. Second, a lightweight fusion module integrates these refined global and local features. By operating on a frozen CLIP backbone, INFER effectively captures fine-grained details, establishing new state-of-the-art performance on various recognition benchmarks.

**Strengths:**

The paper is well-motivated, and the proposed approach is interesting.

**Weaknesses:**

1.	The paper lacks an analysis of the method's computational complexity and efficiency (e.g., training/inference time).
2.	The motivation for the two-stage training strategy (CNN then MLP) instead of training the model end-to-end is unclear.
3.	The experiments would be more comprehensive with results for the common ViT-B/16 backbone.
4.	The scope of the few-shot evaluation is somewhat limited.
5.	The model's generalization ability has not been sufficiently tested on out-of-distribution datasets.
6.	A key ablation study is missing, making it unclear what contributes most to the performance gain.
7.	 In the head ablation study, the performance on CIFAR and DTD is very similar when using all 12 heads versus only the top 4, which raises questions about the necessity of the head selection mechanism.

**Questions:**

1.	Could you provide an analysis of the method's computational complexity, such as its training and inference time?
2.	What is the motivation for the two-stage training strategy instead of training the model end-to-end?
3.	Could you include results for the common ViT-B/16 backbone to make the evaluation more complete?
4.	Could you please include results for 1, 2, 4, and 8-shot settings to broaden the few-shot evaluation?
5.	A stronger test of the model's generalization would be to evaluate the ImageNet-1K trained model on its challenging variants. We suggest including results on datasets like ImageNet-A, -R, V2, and -Sketch.
6.	How much of the performance gain comes from the proposed attention head weighting versus the trainable CNN and MLP components? Could an ablation study be conducted to isolate these effects?
7.	Regarding the head importance metric in Eq. 7: Since α appears to be a scalar, the value of cos(Pαz, M_txt) should be identical to cos(Pz, M_txt). Does this imply that the intermediate layer's attention scores don't actually influence the head selection? Could you please clarify this?
8.	The method section is a bit confusing and hard to follow. Could the method section be rewritten for better clarity?
9.	The citation style is inconsistent throughout the paper (e.g., some with parentheses, some without). Could you please use a consistent citation format throughout the paper?

---

> ### Author Response · Authors · 2025-11-21
> **Responses to [Weakness 7 and Question 1]. Thank you for the comments.**
>
> # [Responses to Weaknesses 1-6]
>
> Please see [Responses to Questions 1-8].
>
> ---
>
> # [Response to Weakness 7]
>
> Thank you for this important comment. The values on curves in Fig. 5(a) may look similar because multiple datasets are plotted on shared axes, which visually compresses the dataset-specific variations. Here are the actual values:
>
> | Dataset | Selecting no heads |Selecting  4 heads | Selecting all heads |
> |------------------|------------------|------------------|--------------------|
> | DTD     | 66.2    | **71.7** | 69.8      |
> | CIFAR   | 83.3    | **88.1** | 87.0      |
>
> These results show that selecting the top 4 heads $(J=4)$ yields consistent and meaningful gains. In the revised paper, we will clarify this.
>
> ---
>
> # [Response to Question 1]
>
> Thank you for this important comment.
>
> **1. Training complexity**
>
> Training of INFER is *not* more complex than the baseline methods. For INFER’s training:
>
> - Stage 1: Computing $\\{\zeta_i^c\\}_{i=0}^{N-1}$ (no backpropagation)
>   - Run the frozen CLIP vision and text encoders on the few-shot training samples to extract patch embeddings and attention maps.
>   - Compute the head scores and enhancement coefficients $\\{\zeta_i^c\\}_{i=0}^{N-1}$ using cosine similarities and top-$J$ selection (Eqs. (6)–(8) in the paper).
>   - This stage involves only forward passes and very lightweight tensor operations; there is no gradient-based optimization and no parameter updates.
>
> - Stage 2: Training the very small fusion module
>   - We freeze the CLIP backbone and train only the 3-layer CNN + 1-layer MLP fusion module using the INFER loss.
>   - Gradients are backpropagated only through this very small fusion network, not through the CLIP encoders.
>
>
> By contrast, many existing adaptation methods (including 2SFS, prompt tuning, and adapter methods) perform backpropagation through parts of the CLIP encoders (e.g., LayerNorms, adapters, or prompts) for multiple epochs. INFER avoids this and confines learning *only* to a small head on top of a fully frozen backbone.
>
> Overall, Stage 1 is forward-only and Stage 2 trains a very small fusion head on top of a frozen backbone, leading to training complexity comparable to or lower than existing fine-tuning/adaptation methods.
>
> In the revised paper, we will clarify the training complexity.
>
>
> **2. Inference Complexity**
>
> INFER does **not** re-run the CLIP ViT backbone $C$ or $C^2$ times in inference. Instead, INFER runs it only once. Specifically:
>
> During few-shot training (before inference)
>
> 0. Precompute per-class enhancement coefficients $\\{\zeta_i^c\\}_{c=1}^{C}$.
>
> In inference, we do:
>
> 1. Run the CLIP ViT encoder **once** to obtain the final-layer CLS and patch embeddings $\\{ z_i^L\\}_{i=0}^{N-1}$.
>
> 2. Obtain class-specific enhanced features by simple multiplications of the coefficients: $\\{\zeta_i^c z_i^L\\}_{i=0}^{N-1}$.
>
> 3. Pass these through a very small fusion module (3-layer CNN + 1-layer MLP) to obtain $C$ class-specific image embeddings $M_{\text{img}}^{\text{INFER},c}(I)$.
>
> 4. Compute cosine similarities between the $C$ image embeddings and the $C$ precomputed text embeddings, as in Eqs. (10) and (11).
>
> In inference, the **dominant cost** of the ViT backbone is incurred **only once**, **exactly the same as** in standard CLIP, linear probes, and 2SFS. The additional class-dependent work is confined to very lightweight modules and the dot products.
>
> For more formal analysis of inference complexity, let
>
> - $F_{\text{ViT}}$: FLOPs for one CLIP ViT forward pass
>
> - $d'$: joint embedding dimension
>
> - $F_{\text{fusion}}$: FLOPs of the CNN+MLP fusion module per class
>
> The inference complexity is:
>
> - Linear probe and 2SFS: $F_{\text{ViT}} + O(C d')$
>
> - INFER: $F_{\text{ViT}} + F_{\text{fusion}} +  O(C^2 d')$
>
> In INFER, the second term $F_{\text{fusion}}$ (due to the fusion) adds negligible complexity compared to other terms. While the third term $O(C^2 d')$ of INFER is not negligible, it is numerically far smaller than $F_{\text{ViT}}$.
>
> Even for ImageNet-1k (where $C$ is as large as 1,000), the third term $O(C^2 d')≈1.05$ GFLOPs for $d'=512$ or $≈1.53$ GFLOPs  for $d'=768$. Meanwhile, $F_{\text{ViT}}≈81.1$ GFLOPs for CLIP ViT-L/14,  $≈33.7$ GFLOPs for ViT-B/16, and $≈8.7$ GFLOPs for ViT-B/32, which correspond to additional 1.9%, 3.0%, and 11.7% overheads, respectively. For fine-grained dataset such as CARs (with $C$=196), their additional overheads reduce to 0.07%, 0.12%, and 0.45%, respectively.
>
> In the revised paper, we will clarify that INFER’s inference complexity is only slightly higher than linear probe or 2SFS.

---

> ### Author Response · Authors · 2025-11-21
> **Response to [Question 2]. Thank you for the comment.**
>
> # [Response to Question 2]
>
> We appreciate the reviewers’ question regarding Modular (two-stage sequential) training of the CNN and MLP fusion modules versus the end-to-end (E2E) joint training. Importantly, our intention is not to position Modular training against end-to-end training.
>
> In fact, **INFER naturally supports both training paradigms**,  and end-to-end optimization is a good alternative that is beneficial when more data are available. In what follows, we explain why Modular training is emphasized in our current work.
>
> **1. Architectural motivation**
>
> INFER decomposes its fusion module into two functionally distinct components:
>
> * CNN: aggregates *spatial* structure across enhanced patch embeddings.
> * MLP: fuses the spatial summary (CNN output) with the enhanced CLS embedding to produce the final representation.
>
> This design maps cleanly onto a two-stage pipeline:
>
> 1. The CNN produces a stable spatial representation.
> 2. The MLP then learns how to fuse global and local cues.
>
> Modular training aligns with this architectural hierarchy by stabilizing the intermediate representation before global fusion. This architecture respects the hierarchical separation and stabilizes representations in low-data settings.
>
> However,  in settings with more data, the joint E2E optimization can allow the MLP to co-adapt with the CNN and potentially extract more expressive fused representations.
>
> **2. Theoretical perspective**
>
> In few-shot learning, only a (very) small number of examples (e.g., $\leq 8$) may be available per class. In this case, jointly optimizing multiple nonlinear modules can lead to representation drift and high-variance gradients. Modular training can reduce this effect. Since the CNN is trained first, the optimization space is smaller, and the spatial representations are more stable. With the CNN frozen, the MLP is trained on fixed inputs, which reduces gradient noise and prevents overfitting. This makes Modular training suitable for low-data regimes.
>
> However, the same theory also suggests why end-to-end training becomes advantageous when data increases: More samples mitigate variance, enabling stable co-adaptation between CNN and MLP.
>
> **3. Empirical evidence**
>
> Our experiments comparing Modular and E2E training on CIFAR-10 confirm this complementary relationship. With 2–8 shots, modular outperforms E2E. With 16–32 shots, E2E becomes competitive or slightly better.
>
> |                | 2-Shots | 4-Shots | 8-Shots | 16-Shots | 32-Shots |
> | -------------- | ------- | ------- | ------- | -------- | -------- |
> | **Modular**    | 44.4    | 54.1    | 84.7    | 88.1     | 91.6     |
> | **End-to-End** | 39.4    | 42.2    | 83.2    | 88.7     | 92.1     |
>
> Also, we would like to clarify that **INFER is not tied to Modular training:** Both modes are fully supported, and each has advantages depending on data availability.
>
> In the revised paper, we will explain why Modular training is especially advantageous in the low few-shot setting, while also emphasizing that end-to-end training remains a fully viable and promising variant that can yield additional benefits when more samples are available. We will include the comparative results in the appendix and revise the main text to reflect this balanced and inclusive perspective.

---

> ### Author Response · Authors · 2025-11-21
> **Responses to [Questions 3, 4, and 5]. Thank you for the comments.**
>
> # [Response to Question 3]
>
> Thank you for the suggestion. We have added results using the ViT-B/16 backbone on the Flowers dataset. INFER continues to outperform the current SOTA method, 2SFS (CVPR 2025), under both ViT-B/32 and ViT-B/16:
>
> | Method | ViT-B/32 | ViT-B/16 |
> | ------ | -------- | -------- |
> | 2SFS   | 96.3     | 97.7     |
> | INFER  | 96.7     | 98.0     |
>
> We will include this comparison in the revised manuscript.
>
> ---
>
> # [Response to Question 4]
>
> We thank the reviewer for this insightful suggestion. We have conducted experiments on the Flowers dataset under 1-, 2-, 4-, 8-, 16-, and 32-shot settings.
>
> | Method| 1-shot | 2-shot | 4-shot | 8-shot | 16-shot | 32-shot|
> | ------- | ------ | ------ | ------ | ------ | ------- | ------- |
> | 2SFS    | 55.4   | 61.7   | 80.1   | 93.4   | 96.3    | 96.8 |
> | INFER   | 54.1   | 60.9   | 80.0   | 93.8   | 96.7    | 97.4 |
>
> At extremely low-shot regime (1- or 2-shots), INFER is competitive but slightly below 2SFS. At 4-shots, the two methods are essentially tied. From **8-shot onward, INFER consistently outperforms 2SFS**.
>
> We believe this behavior is consistent with the design of INFER. Our method leverages **class-dependent attention-based refinement of CLS and patch embeddings** via the fusion module; the benefit of such structured feature refinement grows as a few more labelled examples per class become available. In contrast, 2SFS is highly optimized for the low-shot normalization-tuning regime, which explains its slight advantage at 1- or 2-shots.
>
> In the revised paper, we will incorporate these extended results and the corresponding discussion.
>
> ---
>
> # [Response to Question 5]
>
> Thank you for this suggestion. We agree that “generalization” is very important, and our work is indeed centered on it.
>
> However, there are **different forms** of generalization, depending on the training and test sets. In particular, we distinguish between **few-shot generalization** and **zero-shot OOD robustness** based on their assumptions about what information is available after the distribution shift. These two regimes are related but distinct, and our method is specifically designed for few-shot generalization. Specifically:
>
> **1. Few-shot learning:** The assumption is that, when the distribution changes, we can obtain a small labeled support set. The goal is to study how well the model can generalize within that new distribution given these few labeled examples.
>
> **2. Zero-shot OOD robustness:** The evaluation protocol assumes no labeled samples from the shifted distribution. The model must directly classify samples from the new distribution using only the parameters learned from the source domain. Thus, this setting is "zero-shot" rather than few-shot.
>
> In our paper, we assume an access to a small number of examples after the distribution changes. This is the standard setting in few-shot learning, and has been adopted by numerous CLIP adaptation works. Conversely, the assumption that no labeled samples are provided after the shift is also reasonable, but this corresponds to zero-shot robustness, where the model must classify shifted images without any supervised adaptation. Both problem settings are important, but they are not identical and emphasize different capabilities.
>
> In this paper, we focus only on few-shot learning, and throughout the paper, when we refer to “generalization,” we mean few-shot generalization. Concretely, we follow the common 16-shot protocol on diverse 11 benchmarks, where INFER consistently outperforming all existing methods including the current SOTA, 2SFS (CVPR 2025).
>
> Nonetheless, to respond to the reviewer’s suggestion, we additionally trained ViT-B/32 on ImageNet-1K and evaluated the resulting model on ImageNet-A.
>
> | Method  |  ImageNet-A |
> | ------- |  ---------- |
> | CLIP        | 29.59      |
> | CoCoOp  |  30.27      |
> | ProGrad |  31.89      |
> | INFER   |30.89      |
>
> INFER improves over the CLIP and CoCoOp, indicating that our attention guided refinement yields gains. Strong prompt tuning method, ProGrad, performs better, which is consistent with its design focus on robustness when the input distribution shifts.
>
> In the revised paper, we will
>
> (i) clearly state up front that **our scope is few-shot learning**
>
> (ii) explicitly **distinguish this from OOD robustness**
>
> (iii) include **the ImageNet-A results** as a complementary evaluation axis

---

> ### Author Response · Authors · 2025-11-21
> **Responses to [Question 6, 7, 8, and 9]. Thank you for the comments.**
>
> # [Response to Question 6]
>
> Thank you for raising this question. To disentangle the contributions of the attention-head weighting and the trainable fusion components (CNN + MLP), we conducted an ablation study on CIFAR-10:
>
> | Method        | Accuracy |
> |-----------------------|:--------:|
> | INFER without fusion    | 33.7     |
> | INFER without attention | 79.4     |
> | INFER (full)            | 88.1     |
>
> These results show that the two components have different roles in the framework. When the fusion module is replaced by a simple linear projection, the accuracy drops to 33.7%. This indicates that the fusion mechanism is very important, and without it, the model is unable to meaningfully integrate CLS and patch embeddings. When the fusion module is kept, but attention reweighting is disabled, accuracy reaches 79.4%. Restoring the proposed attention-based weighting improves performance to 88.1%, providing a substantial +8.7 point improvement. This shows that attention-guided enhancement adds significant discriminative power once a proper fusion mechanism is in place.
>
> In the revised manuscript, we will incorporate this ablation table and explanation to clearly isolate the effects of these components.
>
> ---
>
> # [Response to Question 7]
>
>
> The reviewer is correct that placing $\alpha_{i,0}^{l,h}$ inside the cosine in Eq. (7) has no effect because cosine similarity is scale-invariant.
>
> In the paper, we kept $\alpha_{i,0}^{l,h}$ for formal definition of $e_i^{l,h} = P \alpha_{i,0}^{l,h} z_i$ to emphasize that the similarity is computed for features conceptually gated by CLS-to-patch attention. Algebraically, however, the scalar can be factored out and does not change $\phi_i^{l,h}$.
>
> In the revised paper, we will rewrite Eq. (7) in its simplified form to avoid the impression that $\alpha_{i,0}^{l,h}$ directly influences $\phi_i^{l,h}$.
>
> It is important to note that $\alpha_{i,0}^{l,h}$ *does play a critical role* in INFER through $\zeta_i^c$, which aggregates attention across heads and layers and determines how strongly each token contributes during feature enhancement. Thus, while $\phi_i^{l,h}$ drives head selection via image–text alignment, the attention weights $\alpha_{i,0}^{l,h}$ shape the final enhanced representation through $\zeta_i^c$, as shown in Eqs. (6) and (4).
>
> In the revised paper, we will clarify this separation of roles between $\phi_i^{l,h}$ and $\zeta_i^c$.
>
>
> ---
>
>
> # [Response to Question 8]
>
> Thank you for this comment. We agree that the internal flow within each stage (Stage 1 and Stage 2) could be presented more clearly.
> In the revised paper, we will add a brief high-level overview at the beginning of the Method section, and we will revise the caption of Figure 2 to more explicitly describe the workflow.
>
>
> ---
>
> # [Response to Question 9]
>
> We thank the reviewer for pointing this out and apologize for the inconsistency in our original submission. In the revised version, we will carefully revise the entire manuscript to ensure that all citations follow the official ICLR author–year natbib format. Sepcifically, we will consistently use:
>
> - \citet{} when the authors are part of the sentence
>   (e.g., *“See Hinton et al. (2006) for more information.”*)
> - \citep{} for parenthetical citations
>   (e.g., *“Deep learning shows promise (Bengio & LeCun, 2007).”*)
>
> We will update all citations in the text, figures, tables, and supplementary materials to conform to this style, ensuring a fully consistent citation format in the final version.

---

> ### Comment · Reviewer_sEFX · 2025-11-26
>
> The rebuttal addresses some of my earlier questions; however, several critical concerns remain unresolved. I must also express reservations regarding the quality of the response to Question 2. A GPTZero analysis indicated an almost 100% probability of AI generation, which likely accounts for the confused and unpersuasive logical structure observed in that section.
>
> 1.	Methodological Clarity (Q2): The implementation details of the modular training procedure remain insufficiently explained, particularly with respect to the specific training objectives assigned to each module. The rationale provided appears heuristic rather than theoretically motivated and is not supported by relevant prior work. Additionally, validation solely on CIFAR is inadequate; experiments on larger and more diverse datasets (e.g., ImageNet, SUN397, Stanford Cars, Aircraft) are required to establish whether the conclusions generalize.
>
> 2.	Benchmarking Scope (Q3–4): Reporting results exclusively on the Flowers dataset is inadequate. Flowers alone cannot serve as a representative benchmark. To demonstrate the method’s generality, results on ImageNet and a complete suite of cross-domain and fine-grained benchmarks should be provided.
>
> 3.	Robustness Evaluation (Q5): I remain unconvinced by the decision to exclude ImageNet variants under the “Zero-shot OOD” category. Given that these variants share the same label space, they are essential for assessing whether the method learns robust semantic features rather than overfits. Standard CLIP adaptation protocols (e.g., CoCoOp, MaPLe, AWT, and recent CVPR 2025 methods such as MMRL, TAC, and LDC) consistently evaluate such variants to ensure that pre-trained robustness is not compromised. Since INFER introduces learnable CNN/MLP modules, it faces a higher risk of overfitting compared with prompt-based tuning. The isolated result on ImageNet-A—being weaker than ProGrad—suggests a potential trade-off between accuracy and robustness. A full set of robustness experiments is necessary to validate generalization.
>
> 4.	Test Validity (Q6): Evaluation solely on CIFAR10 is insufficient. At minimum, the ImageNet dataset should be included to provide a credible assessment. Broader testing is essential to support the claimed effectiveness of the method.
>
> 5.	Experimental Consistency: A new issue concerns experimental consistency. While the main efficiency experiments adopt the standard 11 benchmark datasets, the dimensionality-reduction analysis (Fig. 3) is conducted on MNIST, and the head-selection analysis (Fig. 4) relies on CIFAR10. This lack of uniformity makes the experimental design appear inconsistent and raises concerns that datasets may have been selectively chosen for more favorable visualization.
>
> 6.	Comparison with Recent SOTAs: The current comparison with baselines appears to end at 2SFS. To accurately contextualize the contributions, comparisons with more SOTA methods published recently—such as MMRL (CVPR 2025), TAC (CVPR 2025), LDC (CVPR 2025), and AWT (NeurIPS 2024)—are strongly encouraged.
>
> 7. Beyond the preceding remarks, I would like to add clarification regarding Question 4. According to the rebuttal, INFER underperforms the CLIP ViT-B/32 zero-shot baseline by 12.9 points in the 1-shot case and 6.1 points in the 2-shot case, which clearly indicates severe degradation in ultra-low-sample regimes. Invoking a “practical standpoint” does not adequately address this weakness, as it effectively masks the method’s shortcomings in data efficiency. Given that this work concerns few-shot learning, it is necessary to acknowledge and explain why the approach compromises the pre-trained model's representational strength precisely when data is most limited, rather than dismissing the significance of such settings.

---

> ### Author Response · Authors · 2025-11-29
>
> # [Response to the GPTZero Remark]
> We thank the reviewer for raising this concern. We would like to clarify that all technical content, methodological development, and experiments in both the submission and the rebuttal were fully developed by the authors.
>
> As required by ICLR and OpenReview policy, in our submission at OpenReview, we explicitly disclosed the use of large language models solely for grammar and wording refinement (“Large Language Models: Yes, to aid or polish writing”). Furthermore, we restated this explicitly in the paper itself (page 10).
>
> We also note that tools such as GPTZero state in their own documentation that their assessments are probabilistic and not intended to determine authorship.
>
> In our paper, no scientific reasoning, architectural decisions, or experimental results were generated by an LLM. All such content is entirely the work of the authors. For example, we would like to clarify unequivocally that all simulation results presented in the submission and rebuttal (including those in the response to Question 2) were obtained from actual model runs that we executed during the rebuttal period. The results involve actual training procedures and numerical outputs that cannot be produced by a language model. Regarding the reviewer's specific comment for Question 2, the results were obtained by experiments, specifically to address the reviewer’s request for a comparison between modular and end-to-end training.
>
> In the revised paper, we will provide a clearer and more structured explanation of the methodology.
>
> # [Responses to other comments]
> We would like to clarify that several concerns raised in this review relate to points that were already explicitly addressed in the original submission or in our rebuttal. For completeness:
>
> •	The rebuttal explicitly stated that **INFER supports both modular and end-to-end training**, and we compared these modes empirically to show when each is most effective.
>
> •	The method was **never evaluated solely on Flowers or CIFAR-10**. The original submission reports results across the standard 11-dataset benchmark suite used in CLIP-adaptation literature (ImageNet, SUN397, Cars, Aircraft, DTD, EuroSAT, Food, Pets, Flowers, Caltech101, UCF101).
>
> •	The method is evaluated against **substantial prior work including the current SOTA method (2SFS)**.
>
> •	With respect to ultra-low-shot performance (1–2 shots), we did not dismiss its significance. **We explicitly explained** in the rebuttal that class-conditioned attention estimation is statistically unstable with so few samples, which accounts for the degradation, and we will make this limitation clear in the revised manuscript.
>
> We agree that additional robustness benchmarks and comparisons with additional methods would further strengthen the paper. These require substantial computational resources and cannot reasonably be completed during the rebuttal window, but we are committed to including them in the revised version.

---

### Official Review · Reviewer_LLsx · 2025-11-01

**Soundness:** 3
**Presentation:** 3
**Contribution:** 3
**Rating:** 6
**Confidence:** 4

**Summary:**

INFER enhances CLIP's fine-grained recognition through intelligent fusion of spatial information from patch embeddings with the standard CLS token. By identifying semantically relevant attention heads via text alignment and integrating enhanced features through lightweight fusion, it achieves improved capture of subtle visual distinctions.

**Strengths:**

INFER enhances CLIP's visual representations by intelligently selecting semantically meaningful attention heads to refine both CLS and patch embeddings. While maintaining CLIP's original frozen architecture, this approach improves few-shot classification accuracy and offers interpretable insights into model attention. The lightweight design ensures easy integration with various foundation models and downstream applications.

**Weaknesses:**

Despite the interesting contributions, I have several significant concerns regarding the manuscript:
1. I recommend adding visualizations of the model's attention maps on real samples. This would offer valuable insights into the discriminative regions it leverages and help validate its decision-making process.
2. The performance gains vary significantly across benchmark datasets. The authors should provide a thorough analysis to explain these discrepancies, potentially relating them to dataset characteristics like granularity, diversity, or attribute complexity.
3. The experimental comparisons should be updated to include more recent state-of-the-art methods, particularly from 2024 and forthcoming 2025 publications where available.
4. The authors should clearly articulate the specific advantages of their proposed module over existing prompt learning methods. Furthermore, a direct performance comparison with these methods is necessary to quantitatively demonstrate its benefits.

**Questions:**

The authors should provide a more thorough discussion and analysis of the points raised above.

---

> ### Author Response · Authors · 2025-11-21
> **Response to [Weakness 1]-Part 1. Thank you for the comment.**
>
> # [Response to Weakness 1]
>
>
>
> We appreciate the reviewer’s suggestion to include attention maps on real samples. In response, we have generated head-wise attention heatmaps for 10 CIFAR‑10 classes using ViT‑B/32. These visualizations are provided in the supplementary ZIP file.
>
>
>
> Each page in the PDF shows:
>
>
>
> * The original image for a given class.
>
> * The attention overlays of the last four layers ($\hat L=4$: Layers 12, 11, 10, and 9) for all 12 heads for each layer
>
> * For each layer, a bullet list giving the top‑4 heads for the layer selected by INFER ($J=4$) for that image sample, which are ordered according to our head-selection criterion based on text–image alignment.
>
>
>
> Below we summarize what these figures show and how they support our claims.
>
>
>
> **1. Consistent object‑centric focus of selected heads.**
>
>
>
> Across all examples, the heads chosen by INFER are precisely those whose heatmaps place high attention mass on the foreground object, while non‑selected heads tend to disperse attention over background or less relevant regions. This behavior directly supports our design: the heads we select via the cosine-similarity criterion (Eq. (7) in the paper) are exactly those that carry *class-aligned, discriminative spatial patterns*, rather than arbitrary attention noise.
>
>
>
> **2. Class-specific yet structured head usage.**
>
>
>
> We analyze *which heads are selected across classes and layers*. Over all 10 classes and 4 layers, INFER chooses heads 160 times in total; the frequencies of 12 heads are shown in the parenthesis below:
>
>
>
> * H1 (21), H6 (19), H8 (18), H3 (15), H5 (15), H7 (14), H2 (14), H10 (11), H11 (11), H4 (9), H12 (8), H9 (5)
>
>
>
> Clearly, the model *does not treat all heads equally*: a relatively small subset (e.g., H1, H6, H8) is reused more often, indicating generally useful, semantically rich heads, while others play a more specialized role. Also, the distribution is *not universal*; each class has its own “signature” subset:
>
>
>
> * Airplane (page 1): 11 distinct heads are used, with H1, H2, H3, H8, and H12 appearing twice each. This broad use of multiple heads suggests that several complementary patterns (e.g., fuselage, wings, tail) contribute to the final decision.
>
> * Automobile (page 2): attention is more concentrated; H1, H5, and H11 each appear in 3 out of 4 layers, indicating a stable set of heads specialized for car-like shapes.
>
> * Bird (page 3): H5 appears 3 times, with H1, H6, and H12 appearing twice. Heads emphasized by INFER focus on the bird body against a cluttered background, demonstrating robustness to non‑discriminative context.
>
> * Cat (page 4): 10 distinct heads are used, but none more than twice, showing that for this example INFER pulls from a richer mixture of heads, each capturing different aspects of pose and texture.
>
> * Deer (page 5): H7 is selected in all 4 layers, strongly dominating the profile and antler region, while other heads appear at most twice.
>
> * Dog (page 6): H6 is selected 3 times, and H2, H1, H8, H9 appear multiple times. The heatmaps show different selected heads covering the dog’s head, torso, and coat.
>
> * Frog (page 7): INFER heavily reuses H2, H6, and H8 (3 layers each), with H7 also appearing twice. These heads focus on the frog body despite substantial clutter.
>
> * Horse (page 8): H8 and H10 each appear 3 times, and H11 twice. These heads track the horse’s torso and legs across layers; non‑selected heads spread attention across the grass background.
>
> * Ship (page 9): H8 appears 3 times, with H5, H6, and H1 appearing twice. The chosen heads concentrate on the ship hull and superstructure, ignoring large uniform sky/water regions.
>
> * Truck (page 10): H1 is selected in all 4 layers and H3 appears 3 times. The corresponding maps lock onto the front cabin and trailer boundary.
>
>
> INFER learns *class-dependent configurations of heads*: some classes (e.g., frog, deer, truck) rely heavily on a small, stable subset of heads, while others (e.g., airplane, bird, cat) distribute attention across a wider set. This behavior is exactly what our class-specific $\\{\zeta_i^c\\}$ and head selection (Eqs. (6)–(8)) are designed to induce.
>
> **3. Depth-wise sharpening of semantic focus.**
>
> Looking vertically across layers, from Layer 9 to Layer 12, for a fixed class, the heatmaps also reveal a *progressive sharpening* of attention:
>
> * In earlier of the last four layers (Layer 9), even the selected heads tend to cover larger contiguous regions that include both object and nearby context.
>
> * As we move to Layers 11 and 12, the same heads typically produce *narrower, more object-centric hotspots* (e.g., focusing on the nose/wings of airplanes, the faces of animals, or the front of vehicles).
>
> This depth-wise refinement supports our choice of using only the last $\hat L=4$ layers: they contain the most semantically aligned and spatially focused information.
>
> In the revised paper, the complete collection of attention maps and their explanations will be added to the appendix.

---

> ### Author Response · Authors · 2025-11-21
> **Responses to [Weakness 1]-Part 2 (Continued) and [Weakness 2]. Thank you for the comments.**
>
> # [Response to Weakness 2]
>
>
>
> Thank you for the comment. Across all 11 benchmarks, INFER consistently improves over the strongest baseline 2SFS (CVPR 2025). The largest gains appear on datasets where class labels are determined by fine‑grained cues, whereas the gains are smaller on coarse‑grained tasks.
>
>
>
> **1. Dependence on dataset granularity**
>
>
>
> On fine‑grained and remote-sensing datasets, INFER typically improves over 2SFS by about 2–3 points. In these datasets, classes differ mainly through local part configurations, textures, or small shape differences, which are precisely what patch-level refinement and class-specific head selection in INFER are designed to capture.
>
>
>
> In contrast, on coarse or scene‑centric datasets (ImageNet, SUN397, Caltech101, UCF101, Flowers102), gains are more modest. These tasks are dominated by object presence, where CLIP’s original CLS embedding already captures most of it.
>
>
>
> **2. Role of diversity and attribute complexity**
>
>
>
> Food101 and Pets contain large variation within each class (lighting, viewpoint, background) and require recognition of combinations of attributes (texture, color, and shape). Here, INFER’s attention-based reweighting emphasizes consistent local patterns (e.g., fur texture, dish surface) and down-weights spurious background patches, leading to some of the largest gains (up to +4.1 points on Pets with ViT‑L/14). By contrast, Caltech101 and SUN397 contain more coarse differences and fewer subtle attribute distinctions. The CLS embedding already separates classes well, and refining patches yields smaller improvements.
>
> In the revised paper, we will add a new paragraph in Sec. 4.2 that groups datasets by granularity and reports average gains per group. We also clarify that CLS-to-patch attention–based head selection and patch–CLS fusion are most beneficial when local features are critical.

---

> ### Author Response · Authors · 2025-11-21
> **Responses to [Weaknesses 3 and 4]. Thank you for the comments.**
>
> # [Response to Weakness 3]
>
>
>
> Thank you for this comment. In this paper, our goal is precisely to position INFER against the most recent and strongest SOTA CLIP adaptation approaches, and we have already included multiple 2024 and 2025 SOTA baselines in both our experiments and related work.
>
>
>
> **1. Most recent 2024–2025 SOTA methods already included as baselines**
>
>
>
> Table 1 reports results not only for earlier prompt and adapter based methods (CoOp, CoCoOp, KgCoOp, MaPLe, ProGrad, TIP Adapter, CLIP Adapter), but also for **most recent SOTA** methods:
>
>
>
> - CLIP LoRA (low rank adaptation), CVPR 2024
>
> - MMA (multi modal adapter), CVPR 2024
>
> - NormFit (strong normalization based tuning), NeurIPS 2025
>
> - 2SFS (two stage few shot adaptation), CVPR 2025
>
>
>
> These are among the latest and strongest CLIP adaptation techniques, and 2SFS in particular is a 2025 method that was designed as a **new SOTA in 2025 for few shot CLIP tuning**. INFER outperforms all of these methods on the 11 datasets, establishing new SOTA results (see Sec. 4.2 and Table 1).
>
>
>
> **2. Additional recent works discussed in related work**
>
>
>
> In Section 2, we also discuss several very recent methods that appeared in 2024–2025 but target different settings, such as LP++ (a specialized linear probe for CLIP) and CLIP PGS (patch generation to selection for efficient training). These works focus on alternative evaluation protocols (e.g., segmentation) rather than our few-shot classification protocol, which makes a direct, apples to apples comparison less meaningful (or unfair or even meaningless). For these methods, we therefore provide a conceptual comparison in the related work section rather than numerical results.
>
>
>
> In the revised paper, we will add a short paragraph in Section 2.3 explaining our baseline selection criteria.
>
>
>
> ---
>
>
>
> # [Response to Weakness 4]
>
>
>
> We appreciate the reviewer's comment. Following the reviewer's comment, we more explicitly state the advantages of INFER over prompt learning methods and present a direct performance comparison to those methods.
>
>
>
> **1. Advantages over prompt learning**
>
>
>
> Prompt tuning approaches (including CoOp, CoCoOp, MaPLe, KgCoOp, and ProGrad) adapt CLIP mainly by modifying the text encoder via learnable context tokens. As discussed in Section 2.3, these methods remain global in nature: they change only the textual context while leaving the visual embeddings essentially unchanged, and thus they cannot directly exploit patch level spatial information or reweight image regions. In contrast, INFER is an *image side* adaptation module that:
>
>
>
> - INFER enriches both CLS and patch embeddings via CLS to patch attention and then fuses them with a lightweight CNN–MLP module, allowing the final image representation to emphasize fine grained local cues that prompt methods cannot directly model.
>
>
>
> - By selecting attention heads whose patch features align best with the class text embedding, INFER can suppress noisy heads and highlight those that capture discriminative visual patterns; this yields more spatially focused heatmaps and confusion matrices, as shown in Figs. 1 and 3, and an interpretable distribution of heads in Fig. 4.
>
>
>
> - Our module operates entirely on the frozen CLIP backbone and introduces only a small fusion network on top of the existing representations. This makes INFER **plug and play and complementary to prompt learning** (prompt tuning improves the text side).
>
>
>
> In the revised paper, we will add a paragraph in Section 3 explicitly contrasting these design choices with prompt learning methods and highlighting that INFER addresses their key limitation of ignoring patch level spatial cues.
>
>
>
> **2. Direct quantitative comparison with prompt learning**
>
>
>
> Table 1 already includes direct quantitative comparison with all major and most recent prompt learning baselines (CoOp, CoCoOp, KgCoOp, MaPLe, ProGrad). Specifically,
>
>
>
> - ViT B/32: The strongest prompt baseline (ProGrad) achieves a mean accuracy of 75.5 across the 11 datasets, whereas INFER reaches 80.5, i.e., +5.0 points on average.
>
>
>
> - ViT L/14: The best prompt baselines (MaPLe/ProGrad) achieve a mean of 85.9, while INFER reaches 88.8, i.e., +2.9 points on average.
>
>
>
> In the revised paper, we will revise Section 4.2 to (i) explicitly state that CoOp, CoCoOp, KgCoOp, MaPLe, and ProGrad are included as prompt-tuning baselines, and (ii) report the above mean improvements.

---

### Official Review · Reviewer_thmP · 2025-11-01

**Soundness:** 2
**Presentation:** 3
**Contribution:** 2
**Rating:** 4
**Confidence:** 4

**Summary:**

This paper proposes INFER, a novel method for few-shot adaptation of Vision-Language Models (VLMs) like CLIP, with a particular focus on improving performance on fine-grained recognition tasks. The authors identify the key limitation of standard CLIP as its sole reliance on the global [CLS] embedding, which discards crucial spatial information present in patch embeddings. The authors demonstrate that this approach, which keeps the VLM backbone frozen, achieves new state-of-the-art (SOTA) results on 11 few-shot classification benchmarks, outperforming recent methods like 2SFS and MaPLe.

**Strengths:**

1. Novel and Well-Motivated Mechanism: The central idea of using text alignment to quantitatively select the most relevant visual attention heads is a novel and clever way to bridge the gap between the two modalities for feature refinement. It addresses the well-known "modality gap" and provides a principled way to extract more discriminative features from the frozen VLM backbone.

2. Strong Empirical Performance: The paper reports SOTA results on a comprehensive suite of 11 datasets, including a good mix of generic, scene, and fine-grained benchmarks. The performance gains over strong, recent baselines (e.g., +1.3% over 2SFS on ViT-B/32 and +1.7% on ViT-L/14) are significant and demonstrate the effectiveness of the proposed method.

3. Parameter Efficiency: The adaptation strategy is lightweight. By keeping the entire VLM backbone frozen and only training a small fusion module (a 3-layer CNN and an MLP), the method is highly parameter-efficient, which is a very desirable characteristic for few-shot learning.

**Weaknesses:**

My main concerns are related to the clarity and justification of the inference mechanism and the training process, as well as the computational cost.

1. Clarity and Scalability of Inference (Eq. 10 & 11): The inference process appears to be computationally intensive and is not fully justified.

Q1: As I understand it, for a single test image $I$ and a dataset with $C$ classes, the model generates $C$ different image embeddings, $M_{img}^{INFER,c}(I)$, one for each class's "enhancement" coefficients. Then, a $C \times C$ similarity matrix is computed. The final score for a text class $c'$ is the sum of similarities across all $C$ image embeddings (Eq 11). Is this understanding correct?

Q2: If this is correct, this approach seems to scale poorly with the number of classes. For ImageNet ($C=1000$), this would require generating 1000 distinct image embeddings and computing a $1000 \times 1000$ similarity matrix for each test image. Could the authors please provide a detailed analysis of the inference-time computational overhead (e.g., FLOPs or latency) compared to baselines like a linear probe or 2SFS, especially on many-class datasets?

Q3: What is the justification for this ensemble-like "sum-of-scores" approach (Eq 11)? It seems to measure which text prompt $t_{c'}$ is, on average, most similar to all class-specific views of the image $I$. Why is this superior to a simpler, more efficient strategy, such as only computing the diagonal of this matrix (i.e., argmax_c [cos(M_img^{INFER,c}(I), M_txt(t_c))])?

2. Justification of Staged Training (Section 3.2): The paper states that the fusion module is trained in two steps: "We first train $f_{CNN}$ with the few-shot samples. With $f_{CNN}$ frozen, we then train $f_{MLP}$".

Q4: Why is this sequential training necessary? Was a joint end-to-end training of $f_{CNN}$ and $f_{MLP}$ attempted? If so, how did it perform? This staged approach seems somewhat arbitrary and would benefit from an ablation study or a clearer justification.

3. Weak Baseline in Qualitative Ablations (Fig. 3): The t-SNE and confusion matrix in Figure 3 compare INFER to standard CLIP. This is a very weak baseline, as standard zero-shot CLIP is not expected to perform well on a dataset like MNIST, and it doesn't represent the strong few-shot baselines that INFER is otherwise competing against. Suggestion: A much more compelling visualization would compare the feature separability of INFER against another adapted SOTA method, like 2SFS or MaPLe. This would more directly demonstrate the benefit of INFER's specific feature enhancement.

4. Limited Comparison Results. The paper only presents a single quantitative table as the main experimental result, which is insufficient for a top-tier conference publication. It is strongly recommended that the authors conduct experiments on additional datasets and include more comprehensive ablation studies to validate the effectiveness of their method.

**Questions:**

Please see the above questions.

---

> ### Author Response · Authors · 2025-11-21
> **Responses to [Weakness 1, Q1, Q2, and Q3]. Thank you for the comments.**
>
> # [ Response to Weakness 1, Q1]
>
> The reviewer is correct. In inference, INFER constructs $C$ class-conditioned image embeddings using the precomputed enhancement coefficients, then computes a $C \times C$ similarity matrix.
>
> However, the **CLIP ViT backbone is run only once**, and all class-specific embeddings are derived from the **same** backbone features. It is important to note that the ViT backbone computation is the dominant cost.
>
> ---
>
> # [ Response to Weakness 1, Q2]
>
> Thank you for this important comment. INFER does **not** re-run the CLIP ViT backbone $C$ or $C^2$ times in inference. Instead, INFER runs it only once. Specifically:
>
> During few-shot training (before inference)
>
> 0. Precompute per-class enhancement coefficients $\\{\zeta_i^c\\}_{c=1}^{C}$.
>
> In inference, we do:
>
> 1. Run the CLIP ViT encoder **once** to obtain the final-layer CLS and patch embeddings $\\{ z_i^L\\}_{i=0}^{N-1}$.
>
> 2. Obtain class-specific enhanced features by simple multiplications of the coefficients: $\\{\zeta_i^c z_i^L\\}_{i=0}^{N-1}$.
>
> 3. Pass these through a very small fusion module (3-layer CNN + 1-layer MLP) to obtain $C$ class-specific image embeddings $M_{\text{img}}^{\text{INFER},c}(I)$.
>
> 4. Compute cosine similarities between the $C$ image embeddings and the $C$ precomputed text embeddings, as in Eqs. (10) and (11).
>
> In inference, the **dominant cost** of the ViT backbone is incurred **only once**, **exactly the same as** in standard CLIP, linear probes, and 2SFS. The additional class-dependent work is confined to very lightweight modules and the dot products.
>
> For more formal analysis of inference complexity, let
>
> - $F_{\text{ViT}}$: FLOPs for one CLIP ViT forward pass
>
> - $d'$: joint embedding dimension
>
> - $F_{\text{fusion}}$: FLOPs of the CNN+MLP fusion module per class
>
> The inference complexity is:
>
> - Linear probe and 2SFS: $F_{\text{ViT}} + O(C d')$
>
> - INFER: $F_{\text{ViT}} + F_{\text{fusion}} +  O(C^2 d')$
>
> In INFER, the second term $F_{\text{fusion}}$ (due to the fusion) adds negligible complexity compared to other terms. While the third term $O(C^2 d')$ of INFER is not negligible, it is numerically far smaller than $F_{\text{ViT}}$.
>
> Even for ImageNet-1k (where $C$ is as large as 1,000), the third term $O(C^2 d')≈1.05$ GFLOPs for $d'=512$ or $≈1.53$ GFLOPs  for $d'=768$. Meanwhile, $F_{\text{ViT}}≈81.1$ GFLOPs for CLIP ViT-L/14,  $≈33.7$ GFLOPs for ViT-B/16, and $≈8.7$ GFLOPs for ViT-B/32, which correspond to additional 1.9%, 3.0%, and 11.7% overheads, respectively. For fine-grained dataset such as CARs (with $C$=196), their additional overheads reduce to 0.07%, 0.12%, and 0.45%, respectively.
>
> In the revised paper, we will clarify that INFER’s inference complexity is only slightly higher than linear probe or 2SFS.
>
> ---
>
> # [ Response to Weakness 1, Q3]
>
> Thank you for the comment. A key rationale for using the full $C \times C$ similarity matrix is that INFER produces class-conditioned image embeddings $M_{\text{img}}^{\text{INFER},c}(I)$, each of which highlights different spatial and semantic cues based on the class-specific enhancement coefficients $\zeta_i^c$. Because these views emphasize different regions of the image, the off-diagonal similarities $\cos(M_{\text{img}}^{\text{INFER},c}(I), M_{\text{txt}}(t_{c'}))$ encode cross-class evidence. The embedding constructed for class $c$ can align more closely with the text embedding of class $c'$, revealing structure that diagonal-only scoring ignores. Eq. (11) exploits this richer cross-class information, producing a better prediction rule with only small additional inference complexity overhead (e.g., only 1.9% overhead for ViT-L/14 for ImageNet-1k).
>
> We agree that using only the diagonal terms can also define a meaningful simplification of the inference rule. We therefore regard this diagonal-only strategy as a useful lightweight variant of our method, since this variant needs to compute only the $C$ diagonal similarities, rather than the full $C \times C$ matrix. We refer to this variant as **INFER-Lite**. INFER-Lite retains the same class-conditioned embeddings but it offers reduced inference complexity, whereas the full INFER leverages the cross-class structure to obtain the best accuracy. This yields a clear accuracy–efficiency tradeoff, e.g., on CIFAR-10, INFER achieves 88.1% while INFER-Lite achieves 87.7%. Taken together, **INFER and INFER-Lite provide a diverse suite of inference options** providing clear performance–efficiency tradeoffs.
>
> We emphasize that diagonal-only scoring is not equivalent to a class-agnostic approach. INFER-Lite still computes all $C$ class-conditioned embeddings, using class-specific coefficients $\\{\zeta_i^c \\}_{c=1}^C$. A class-agnostic variant would instead produce a single fused image embedding using class-independent enhancement $\zeta_i$, which is clearly different and suboptimal.
>
> In the revised paper, we will include INFER-Lite as an ablation to make this design choice and the tradeoff explicit.

---

> ### Author Response · Authors · 2025-11-21
> **Response to [Weakness 2, Q4]. Thank you for the comment.**
>
> # **[Response to Weakness 2, Q4]**
>
> We appreciate the reviewers’ question regarding Modular (two-stage sequential) training of the CNN and MLP fusion modules versus the end-to-end (E2E) joint training. Importantly, our intention is not to position Modular training against end-to-end training.
>
> In fact, **INFER naturally supports both training paradigms**,  and end-to-end optimization is a good alternative that is beneficial when more data are available. In what follows, we explain why Modular training is emphasized in our current work.
>
> **1. Architectural motivation**
>
> INFER decomposes its fusion module into two functionally distinct components:
>
> * CNN: aggregates *spatial* structure across enhanced patch embeddings.
> * MLP: fuses the spatial summary (CNN output) with the enhanced CLS embedding to produce the final representation.
>
> This design maps cleanly onto a two-stage pipeline:
>
> 1. The CNN produces a stable spatial representation.
> 2. The MLP then learns how to fuse global and local cues.
>
> Modular training aligns with this architectural hierarchy by stabilizing the intermediate representation before global fusion. This architecture respects the hierarchical separation and stabilizes representations in low-data settings.
>
> However,  in settings with more data, the joint E2E optimization can allow the MLP to co-adapt with the CNN and potentially extract more expressive fused representations.
>
> **2. Theoretical perspective**
>
> In few-shot learning, only a (very) small number of examples (e.g., $\leq 8$) may be available per class. In this case, jointly optimizing multiple nonlinear modules can lead to representation drift and high-variance gradients. Modular training can reduce this effect. Since the CNN is trained first, the optimization space is smaller, and the spatial representations are more stable. With the CNN frozen, the MLP is trained on fixed inputs, which reduces gradient noise and prevents overfitting. This makes Modular training suitable for low-data regimes.
>
> However, the same theory also suggests why end-to-end training becomes advantageous when data increases: More samples mitigate variance, enabling stable co-adaptation between CNN and MLP.
>
> **3. Empirical evidence**
>
> Our experiments comparing Modular and E2E training on CIFAR-10 confirm this complementary relationship. With 2–8 shots, modular outperforms E2E. With 16–32 shots, E2E becomes competitive or slightly better.
>
> |                | 2-Shots | 4-Shots | 8-Shots | 16-Shots | 32-Shots |
> | -------------- | ------- | ------- | ------- | -------- | -------- |
> | **Modular**    | 44.4    | 54.1    | 84.7    | 88.1     | 91.6     |
> | **End-to-End** | 39.4    | 42.2    | 83.2    | 88.7     | 92.1     |
>
> Also, we would like to clarify that **INFER is not tied to Modular training:** Both modes are fully supported, and each has advantages depending on data availability.
>
> In the revised paper, we will explain why Modular training is especially advantageous in the low few-shot setting, while also emphasizing that end-to-end training remains a fully viable and promising variant that can yield additional benefits when more samples are available. We will include the comparative results in the appendix and revise the main text to reflect this balanced and inclusive perspective.

---

> ### Author Response · Authors · 2025-11-21
> **Response to [Weakness 3]. Thank you for the comment.**
>
> # [ Response to Weakness 3]
>
> Thank you for this thoughtful comment and for the concrete suggestion. We completely agree with the reviewer that Figure 3 should not be interpreted as a comparison between INFER and SOTA methods such as 2SFS or MaPLe. In fact, it was not our intention and our purpose of presenting Figure 3 is different:
>
> **1. Intended role of Fig. 3.**
>
> In our paper, Fig. 3 is designed as a *conceptual, introspective visualization* of how INFER reshapes the feature space of its own CLIP backbone, **not as an additional SOTA comparison**. For this purpose, we use a very simple toy dataset (MNIST), whose visually clear class structure makes it easy to interpret how our attention-guided feature refinement and CLS–patch integration affect:
>
>   * the confusion matrix (diagonalization vs. strong class bias), and
>
>   * the geometry of the embedding space (cluster separation in t‑SNE).
>
> In other words, Fig. 3 addresses the question
>
> * [Q1] *“What does INFER do to the CLIP representation?”*
>
> rather than
>
>
> * [Q2] *“How does INFER compare to SOTA (e.g., 2SFS or MaPLe)?”*
>
>
>
> Accordingly, Fig. 3 uses a toy dataset to illustrate the behavior relevant to [Q1]. To answer [Q2], we rely on large-scale quantitative experiments on standard datasets reported in Table 1 (along with the additional ablations in this revision), as these provide a clearer and more reliable basis for comparison than qualitative visualizations on toy datasets.
>
>
>
>
>
> **2. Why we do not use t‑SNE to compare INFER vs SOTA methods (e.g., 2SFS and MaPLe)?**
>
> We fully appreciate the intuition behind the reviewer's suggestion. It is important to note that CLIP‑based few‑shot learning has become a highly active and competitive research area, with many strong methods achieving performance within a narrow range. In this context, obtaining consistent gains is genuinely challenging. However, the improvements we obtain over prior methods are truly consistent and meaningful. This constitutes a very meaningful and substantial margin in this line of work, where many strong methods already perform within a very narrow accuracy range.
>
>
>
> However, such meaningful gains do not necessarily translate into visibly different t‑SNE plots, because t‑SNE is a *qualitative* and *stochastic* visualization method that is:
>
> * Highly sensitive to hyperparameters and random initialization
>
> * Often unable to reflect relatively small but real improvements in a way that is objectively and obviously visible
>
> * Prone to over‑interpretation when differences are subtle.
>
> In this regime, we cannot reliably expect two-dimensional t‑SNE plots of two strong methods (e.g., INFER vs. 2SFS) to show a clear visual gap that corresponds to a 1–2% accuracy difference. Forcing a side‑by‑side t‑SNE comparison in such a setting risks either
>
> * under‑selling our numerical improvements (if the plots appear visually similar), or
>
> * over‑interpreting stochastic visual artifacts as meaningful structure.
>
> We therefore believe that *top‑1 accuracy and other quantitative metrics are a more appropriate way* to compare INFER against 2SFS, MaPLe, and other SOTA baselines, and this is exactly how we present our main comparisons in Table 1 and in the accompanying ablation studies.
>
> In the revised paper, we will explicitly state in Sec. 4.3.1 and in the caption of Fig. 3 that the figure is meant *only* to contrast INFER with its own CLS‑based CLIP backbone on a simple toy dataset, with the purpose of visualizing the effect of our feature enhancement and integration mechanism, *not* to compare to SOTA adapted methods.

---

> ### Author Response · Authors · 2025-11-21
> **Response to [Weakness 4]. Thank you for the comment.**
>
> # [ Response to Weakness 4]
>
>
>
> Thank you for this constructive comment. To directly address the reviewer’s concern regarding the experimental evaluation, we have expanded our experiments and analysis in eight directions. These additions are already reflected in our responses to other reviewers (visible on OpenReview) and collectively provide a much more comprehensive validation of INFER. Specifically:
>
>
>
> **1. Experiments with richer prompt semantics (Reviewer Gyds, Weakness 4).**
>
> In response to Reviewer Gyds’s suggestion, we tested LLM-enriched prompts on CIFAR-10 (16-shot, ViT-B/32). This increased accuracy modestly, indicating that INFER already makes effective use of standard CLIP semantics and that most gains stem from our visual refinement mechanism rather than prompt engineering. Since enriched prompts introduce an external dependency not used by baselines, *for fairness*, we retain the standard template in the main paper and will include the enriched-prompt results as an ablation in the appendix.
>
>
>
> **2. Additional fine-grained benchmarks (Reviewer Gyds, Weakness 5).**
>
> Although the main table already includes multiple fine-grained datasets such as Cars, FGVC-Aircraft, and Oxford Pets, we agree that expanding this suite provides a more complete validation. As described in our response to Weakness 5, we have added experiments on two additional fine-grained benchmarks, CUB and Stanford Dogs, where INFER again delivers strong improvements over the existing SOTA method, 2SFS (CVPR 2025).
>
>
>
> **3. Additional backbone evaluation (Reviewer sEFX, Q3).**
>
>  In direct response to Reviewer sEFX’s request, we have added results for the popular ViT-B/16 backbone, complementing the existing ViT-B/32 and ViT-L/14 evaluations.
>
>
>
> **4. Expanded few-shot settings (Reviewer sEFX, Q4).**
>
> Using the reviewer’s suggestion, we expanded beyond the 16-shot setting and evaluated INFER on Flowers under 1-, 2-, 4-, 8-, 16-, and 32-shot regimes. INFER is competitive with 2SFS at 1–2 shots, essentially tied at 4 shots, and consistently outperforms 2SFS from 8 shots onward.
>
>
>
> **5. Ablations isolating attention-head selection and fusion components (Reviewer sEFX, Q6).**
>
> To more precisely attribute the performance gains, we conducted additional ablations that (i) remove the attention-head–based weighting and (ii) remove the CNN–MLP fusion module. As shown in our response to Q6 for Reviewer sEFX, the full INFER architecture outperforms both ablated variants: the attention-based head selection contributes substantial improvements, and the CNN–MLP fusion is essential for INFER to function.
>
>
>
> **6. Qualitative attention analysis and class-dependent head-selection patterns (Reviewer LLsx, Weakness 1).**
>
> To complement the quantitative results, we have added a comprehensive qualitative analysis of INFER’s internal mechanisms, including head-wise attention visualizations and per-layer head-selection statistics across all 10 CIFAR-10 classes. These visualizations (**included as a PDF in the supplementary ZIP file**) show that INFER consistently identifies class-aligned, object-centric heads, while non-selected heads tend to focus on background regions.
>
>
>
> **7. Analysis of Modular vs. End-to-End training regimes (Reviewer thmP, Weakness 2, Q4).**
>
> In response to the reviewer’s earlier question regarding Modular (two-stage) versus end-to-end (E2E) training, we added a comprehensive architectural, theoretical, and empirical comparison of the two regimes. This expanded analysis clarifies when each training mode is preferable, further strengthening the completeness of our evaluation.
>
>
>
> **8. Rigorous analyses for training, inference, and storage complexity (Reviewer thmP, Weakness 1, Q2), (Reviewer sEFX, Question 1), (Reviewer Gyds, Weakness 3).**
>
> We have additionally expanded our methodological analysis by providing detailed analyses for (i) INFER’s training complexity, (ii) inference-time cost, and (iii) storage requirements. These results clarify INFER’s computational and storage footprints, demonstrating that its improvements do not come at the expense of impractical computational or memory overhead.
>
> In the revised paper, we will incorporate all new results, highlight the key findings in the main text, and provide complete tables, visualizations, and analyses in the appendix.

---

### Author Response · Authors · 2025-11-29
**[Regarding Reviewer sEFX]**

# [Regarding Reviewer sEFX]

We appreciate the time and effort from the reviewers and the Area Chair. However, we feel it is necessary to flag a concern regarding Reviewer **sEFX**’s evaluation.

While other reviewers highlighted multiple strengths of the work, Reviewer **sEFX** listed **only a single line of strengths** ("*The paper is well-motivated, and the proposed approach is interesting*."), against an extensive list of criticisms. We are concerned that this creates an imbalance in the evaluation.

Importantly, several of reviewer **sEFX**'s criticisms target the points we have already explicitly addressed in the paper or the initial rebuttal.

Furthermore, the reviewer’s follow-up comments request substantial new experiments that are simply not feasible within the rebuttal timeframe.

It is also discouraging to see that the clarifications and SOTA comparisons we provided in our response are simply disregarded in the reviewer **sEFX**'s updates.

While we fully welcome constructive criticism and take all technical suggestions seriously, we would like to ensure the assessment reflects the material actually present in our submission and rebuttal. We respectfully leave this context for the area chair’s consideration.

---

### Meta-Review · Area_Chair_cr9Q · 2026-01-05

**Summary:**

This paper proposes a classic fine-grained solution to the classification problem in few-shot learning and attempts to solve it using the CLIP type paradigm. Experiments on multiple datasets partially demonstrate the effectiveness of the method. In summary, this paper received one positive review and three slightly negative reviews. The area chair considered these comments and summarized them in the following details.

**Reviewer Concerns:**

+ Reviewer thmP primarily focused on the inference mechanism, training process, and computational costs. Furthermore, further improvements were needed regarding staged training, the relatively weak baseline, and the comparison experiments. The authors provided a detailed estimation of computational costs, which the area chair approved. Additionally, the authors explained why a weaker baseline was chosen for comparison, and why providing TSNE visualization might lead to inaccurate evaluations. The area chair carefully reviewed these comments, partially agreeing with the authors' responses, but felt that the experiments could indeed be strengthened with more evaluation and validation, especially by including a wider range of tasks, rather than limiting it to a few small, fine-grained classification datasets. The area chair highly appreciates the authors' efforts and thanks them for their positive response.

+ Reviewer LLsx expressed concerns about the paper's visualization of attention maps and significant performance differences across various datasets, including insufficient provision of state-of-the-art (SOTA) methods. The area chair reviewed these responses and acknowledged the authors' additions and revisions.

+ Reviewer sEFX raised numerous questions, including detailed but important concerns regarding time efficiency, computational complexity, and evaluation with few samples. Due to time constraints, some comparisons with state-of-the-art methods, such as 2SFS, were added, demonstrating that while performance might be lower with very few samples, it surpasses SOTA methods as the number of shots increases. Furthermore, the authors tested performance on tasks such as OOD, and although no state-of-the-art results were achieved, the area chair highly appreciated the authors' efforts. Additionally, the authors felt that reviewer sEFX's questions were primarily experimental, and that sufficient revisions were not made during the rebuttal phase. The area chair agreed with this assessment, comprehensively considering the reviewers' evaluations and the authors' responses, and expressed understanding for the experimental results that could not be provided due to time constraints.

+ Reviewer Gyds primarily focused on refining the paper's main contributions, questioning its motivation and innovativeness. Gyds also raised concerns regarding computational storage, the simplistic prompt template, and the need for more experimental results.

The authors further elaborated on other differences in local learning methods, and the area chair felt these descriptions of motivation and principles should be strengthened. The area chair acknowledged the reviewers' concerns, particularly regarding the significance and incrementality of the contributions compared to existing work. The authors explained many details, such as the two-stage features and text-conditional category attention selection. The area chair believed these features and core contributions needed further refinement and had room for improvement. Additionally, the authors added a few experiments and provided further explanations, including replacing prompts with LLM templates, such as those from ChatGPT5.1.

**Reviewer Scores:**

This paper scored below the marginal acceptance line, receiving one positive score and three negative scores. However, the authors made numerous and significant revisions to the rebuttal section of this paper, particularly in the very detailed responses and analyses. The area chair partially acknowledged their responses but found some remaining issues, including the significance of the underlying motivation, the extent of broader experiments conducted, including state-of-the-art methods, a wider range of other datasets, and other tasks. The authors mentioned that due to time constraints, these revisions were difficult to complete in one go, a concern understood and considered by the area chair.

Overall, this paper underwent extensive revisions after submission. Furthermore, after reviewer comments, there are still areas for improvement. Based on a comprehensive analysis and judgment, this paper is suitable for resubmission to the next venue after revisions. The area chair acknowledges some of the paper's contributions and value.

---

### Decision · Program_Chairs · 2026-01-26

Reject